# ORIGAMISPACE: Benchmarking Multimodal LLMs in Multi-Step Spatial Reasoning with Mathematical Constraints

**Rui Xu**[1,2,3]     **Dakuan Lu**[3]     **Zicheng Zhao**[1]     **Xiaoyu Tan**[3*]
**Xintao Wang**[1]     **Siyu Yuan**[1]     **Jiangjie Chen**[1]     **Yinghui Xu**[1,3*]
[1]Fudan University     [2]SII     [3]INF Technology
rxu24@m.fudan.edu.cn   xuyinghui@fudan.edu.cn

## Abstract

Spatial reasoning is a key capability in the field of artificial intelligence, especially crucial in areas such as robotics, computer vision, and natural language understanding. However, evaluating the ability of multimodal large language models (MLLMs) in complex spatial reasoning still faces challenges, particularly in scenarios requiring multi-step reasoning and precise mathematical constraints. This paper introduces ORIGAMISPACE, a new dataset and benchmark designed to evaluate the multi-step spatial reasoning ability and the capacity to handle mathematical constraints of MLLMs through origami tasks. The dataset contains 350 data instances, each comprising a strictly formatted crease pattern (CP diagram), the Compiled Flat Pattern, the complete Folding Process, and the final Folded Shape Image. We propose four evaluation tasks: Pattern Prediction, Multi-step Spatial Reasoning, Spatial Relationship Prediction, and End-to-End CP Code Generation. For the CP code generation task, we design an interactive environment and explore the possibility of using reinforcement learning methods to train MLLMs. Through experiments on existing MLLMs, we initially reveal the strengths and weaknesses of these models in handling complex spatial reasoning tasks.

## 1 Introduction

Spatial reasoning is a core component of artificial intelligence [1, 2], with wide applications in robotics [3], autonomous driving [4], and geographic information systems [5]. Although multimodal large language models (MLLMs) demonstrate outstanding performance in various vision and language tasks [6, 7], they face challenges in imagining spatial transformations and grasping spatial relationships in image and text spaces. Evaluating their spatial reasoning ability has become an important task.

Multi-step reasoning and constraints are critical yet underexplored areas in spatial intelligence. Current spatial reasoning benchmarks typically focus on understanding static images or simple scenes [8]. Some studies are dedicated to comparing and reasoning about spatial relationships between image pairs, but lack attention to continuous spatial transformations [9, 10]. Some studies propose multi-step spatial reasoning but do not involve interaction with the environment and lack constraints found in real-world tasks [11]. These limitations indicate a current need for a new benchmark to more comprehensively evaluate the capabilities of MLLMs in complex spatial reasoning scenarios.

Origami art offers an ideal platform for evaluating complex spatial reasoning abilities [12]. Origami involves a sequence of ordered folding operations, where each step depends on the result of the

---

*Corresponding authors.

39th Conference on Neural Information Processing Systems (NeurIPS 2025).

previous one, embodying the essence of multi-step reasoning. Furthermore, the origami process is governed by explicit geometric constraints, such as folds must occur along straight lines, and the paper cannot be torn or separated; all origami operations are defined by strict mathematical constraints (*Kawasaki's Theorem*, *Huzita-Hatori axioms*, etc.) [13, 14]. The transformation from a two-dimensional crease pattern (CP diagram) through multiple folding steps to a three-dimensional folded shape image requires strong spatial imagination and reasoning abilities.

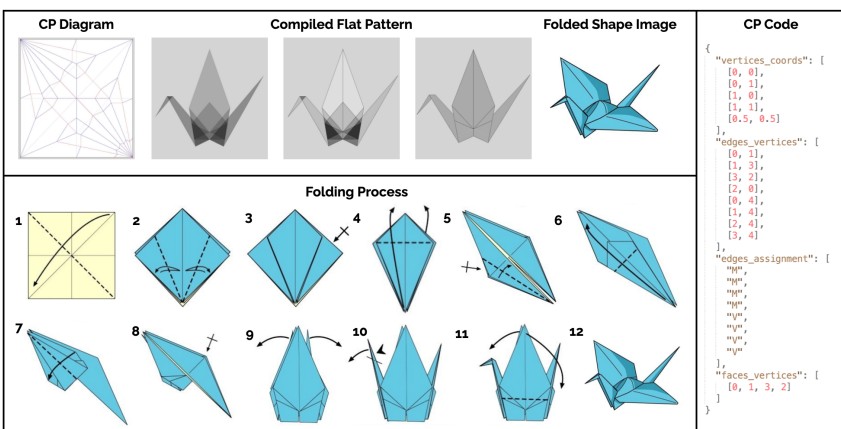

Figure 1: An example data instance from ORIGAMISPACE includes: CP Diagram, Compiled Flat Pattern, Folded Shape Image, and Folding Process, where the CP Diagram can be represented in the form of CP Code.

To bridge the gap of existing benchmarks, this paper introduces the ORIGAMISPACE dataset and benchmark. This dataset contains 350 meticulously collected origami data instances, including a CP diagram, its corresponding compiled flattened pattern, illustrations of the complete folding process, and the final folded shape. The diversity and complexity of the data cover various origami types. We improve the existing origami compiler, enabling it to output detailed flattened diagrams that include crease locations and stacking relationships, support interactive simulation with MLLMs, and provide more comprehensive error feedback. Based on this dataset, we design four challenging evaluation tasks: pattern prediction, spatial relationship prediction, multi-step spatial reasoning, and end-to-end CP code generation, which comprise 1,500 multiple-choice questions and 120 code generation questions. For the code generation task, we meticulously design a comprehensive evaluation strategy to measure the quality of the generated CP code across multiple dimensions.

The core advantages of ORIGAMISPACE lie in its authenticity (derived from real origami designs), multi-step reasoning characteristics (reflecting the inherent process of origami), and rigorous mathematical constraints (precisely verifiable through origami theorems). We evaluate the performance of various MLLMs on ORIGAMISPACE, and introduce environmental learning and reinforcement learning methods for the code generation task, which opens up new perspectives and effective avenues for assessing and enhancing the spatial reasoning abilities of MLLMs.

The main contributions of this paper include:

- We introduce ORIGAMISPACE, a dataset containing 350 high-quality origami data instances, and optimize the existing origami compiler, enabling it to provide more comprehensive feedback.

- We design four challenging tasks centered around spatial reasoning, including 1,500 multiple-choice questions and 120 CP code generation questions, which is the first benchmark to evaluate the multi-step spatial reasoning ability of MLLMs under mathematical constraints.

- We conduct a comprehensive evaluation of existing MLLMs and develop a complete interactive environment for the end-to-end CP code generation task, and explore environmental learning and reinforcement learning methods through this environment.

## 2 Related Work

### 2.1 Spatial Reasoning Benchmarks

Evaluating the spatial reasoning abilities of MLLMs is crucial for advancing their application in real-world scenarios, but existing benchmarks have certain limitations [8, 15]. CLEVR [9] and Visual Genome [16] focus on static scene understanding or single-step reasoning and often operate in synthetic environments, making it challenging to reflect the complexities of the real world. NLVR2 [17] concentrates on comparative reasoning through image pairs but struggles to measure a model's ability to understand and execute tasks involving multiple spatial state transitions. StepGame [18] and LEGO-Puzzles [11] explore multi-step processes, but they are either limited to pure text models or do not sufficiently emphasize precise geometric and physical constraints. Furthermore, interaction with the environment and understanding physical manipulation are also weak points in current evaluation methods. Many benchmarks primarily rely on static inputs and less frequently involve tasks that require models to predict or guide a sequence of physical actions. To address these challenges, we propose ORIGAMISPACE. By introducing origami, a structured and complex multi-step physical task, ORIGAMISPACE directly targets the shortcomings of existing benchmarks. It leverages origami's inherent precise geometric constraints and sequence of operations, aiming to comprehensively and deeply evaluate the capabilities of MLLMs in complex, dynamic spatial reasoning.

### 2.2 Computational Origami

Computational origami is an emerging field within computer science that focuses on studying algorithms for solving origami-related problems [19, 20]. This field covers two main aspects: origami design [20] and origami foldability [21, 22]. Origami design involves the development of algorithms to generate origami crease patterns with specific shapes or functionalities [23]. Origami foldability, on the other hand, investigates how to determine whether a given crease pattern can be folded into a particular shape, especially flat-foldability [24]. Our work does not focus on designing new origami models; instead, we leverage the characteristics of origami to evaluate the spatial reasoning abilities of MLLMs. Drawing upon knowledge from computational origami regarding crease patterns, folding processes, and mathematical principles, we have optimized an existing origami compilation system and evaluation functions for crease patterns, thus establishing a benchmark designed to test the capabilities of MLLMs in multi-step spatial manipulation and constraint satisfaction.

## 3 ORIGAMISPACE Dataset

### 3.1 Data Collection

We collect 350 sets of origami data. These data originate from various online resources, including origami tutorial websites[2,3], forums[4], and origami books[5,6]. As depicted in Figure 1, each complete data entry comprises the following four parts:

**CP Diagram** The CP diagram is a standardized format, representable by code, that displays all the creases of an origami model. It is typically a two-dimensional planar drawing where different line styles indicate different types of folds (e.g., mountain fold, valley fold). Subject to constraints, a CP diagram uniquely determines a folded shape image. The format of the CP diagrams in our dataset adheres to strict requirements, ensuring their correct parsing by our compiler.

**Compiled Flat Pattern** Through the compiler, the final folded state of the CP diagram under all constraints can be computed, and the output compiled flat pattern can represent the two-dimensional state of the origami model after complete folding.

---

[2] https://origami-database.com/

[3] https://github.com/origamimagiro/flat-folder

[4] https://mitani.cs.tsukuba.ac.jp/oripa/

[5] https://www.giladorigami.com/origami-database.php

[6] https://oriwiki.com/

**Folded Shape Image** Different from the strictly compiled flat pattern, the folded shape image provides a direct, intuitive visualization of the final origami shape. It is typically a photograph or 3D rendering.

**Folding Process** The folding process refers to the multi-step sequence of transforming the original paper into the final shape. This folding process is gathered from various origami tutorials and cannot be represented in a standardized format, existing only as natural images.

We manually check and verify all data to ensure that 1) all CP diagrams can be compiled into the compiled flat pattern and correspond to the folded shape image; 2) the names of all origami data correspond to the folded shape image, with no potential for confusion (such as indistinguishable birds); and 3) all folding processes are feasible. In addition to this part of the data, we also collect 471 groups of data without intermediate folding processes for the subsequent training of the model.

## 3.2 Compiler

The current origami compiler computes the final state achievable by a CP diagram under all mathematical constraints, thereby compiling the compiled flat pattern. We have optimized this process: 1) During compilation, we mark each crease, allowing us to locate the position of every crease in the compiled image. 2) We further compute the paper stacking order information, clarifying the top-bottom relationship of different paper regions in the compiled flat pattern. 3) We construct an interface for direct interaction between MLLMs and the compiler, enabling the model to call this system more conveniently to complete origami simulations. 4) We improve the error feedback system of the compiler. Specifically, it returns four types of errors:

**CP Code Syntax Error (CSE)** Validates the existence, format, and validity of inter-references of core data structures in the CP code (such as vertex coordinates `vertices_coords`, edge-vertex relationships `edges_vertices`, and face-vertex relationships `faces_vertices`). It also checks if crease types (e.g., 'B', 'M', 'V', 'F', 'U') are predefined characters, and verifies if *Euler's formula* for planar graphs is satisfied: $V - E + F = 2$, where V, E, and F represent the number of vertices, edges, and faces, respectively.

**Geometrically Impossible Fold (GIF)** Refers to cases where the CP code geometrically violates fundamental origami principles, making the fold physically unrealizable. For example, violating local flat-foldability conditions at a vertex (such as Maekawa's theorem $|M - V| = 2$ or Kawasaki's theorem $\sum \alpha_i = 2\pi$), or specified crease angle combinations would require the paper to be stretched or torn.

**Paper Self-Intersection/Penetration (PSI)** Occurs when logically incompatible situations are found while deducing the relative positions and layering order of different paper sections after folding. This may manifest as a cycle in the calculated paper layering relationships (e.g., layer A is above layer B, layer B is above layer C, and layer C is, in turn, above layer A), or in a 2D unfolded representation, different paper regions are assigned to overlapping positions that would cause physical penetration.

**Ambiguous Folding State (AFS)** This error occurs when a given CP code, due to its inherent under-constrained nature (e.g., allowing multiple valid mountain-valley assignments for creases, or lacking critical information such as crease types or angles), can be compliantly folded into multiple different stable geometric structures, or prevents the compiler from uniquely determining the layering order when processing complex overlapping paper regions.

## 3.3 Dataset Statistics

In ORIGAMISPACE, the distribution of different types of origami is relatively even. To ensure data diversity, we choose origami models covering different levels of complexity and types of folds, such as animals, plants, geometric shapes, etc. The average number of folding steps for origami models is 8.2, but the variation between different models varies greatly, ranging from a minimum of 3 steps to a maximum of 25 steps. Appendix A presents more detailed data analysis, including the themes and names of all origami data and the proportion of different folding steps.

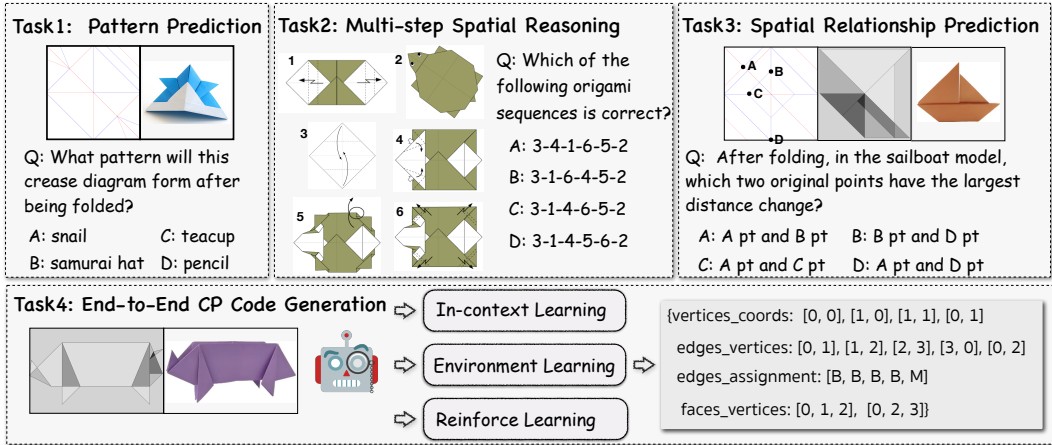

Figure 2: Data examples of the four tasks. The first three tasks are in a multiple-choice format, and the fourth task is a code generation task.

# 4 Task

Based on ORIGAMISPACE, we propose four tasks to evaluate the spatial reasoning capabilities of MLLMs comprehensively.

## 4.1 Pattern Prediction

This task evaluates the model's ability to understand the folding process from the CP diagram and imagine the final 3D shape. For this task, the input is the CP diagram, and MLLMs are required to predict the resulting folded shape image based on it. To enable better quantitative evaluation, we structure this task as a multiple-choice question. The correct option is the name of the target shape. For the incorrect options, three origami enthusiasts design three options for each diagram, adhering to criteria that require them to be easily distinguishable from the correct option; not be variations of the same concept (e.g., if the correct option is a cat, incorrect options are not lions, leopards, etc.); and be close to potential folded states based on the CP diagram (e.g., removing a few key creases makes a boat's CP diagram similar to a hat). We create 350 questions for this task. See Appendix B.1 for the specific annotation rules.

## 4.2 Multi-step Spatial Reasoning

This task evaluates the model's ability to understand the dynamic origami process and the logical relationships between steps. The input for the task is a set of images that collectively show several key steps of a complete origami process. However, the order of these images is randomly shuffled. MLLMs need to infer the correct chronological order in which these steps occur, based on their understanding of the geometric state changes in the images. To better quantify the model's performance, we structure this task as a multiple-choice question. The correct option is the sequence of steps that represents the unique correct folding process (for example, "1-2-3-4"). For the incorrect options, we generate multiple logically incorrect sequences of steps (for example, "1-2-4-3", "4-1-2-3", etc.). These incorrect sequences may contain partially correct local orders but contain errors in the overall flow, in order to test the model's grasp of the complete, coherent process. We design 250 such questions, and the average number of steps per question is 7.5.

## 4.3 Spatial Relationship Prediction

This task evaluates the model's ability to predict spatial relationships and geometric properties after the folding process is complete. For this task, the input is the CP diagram. The model is required to predict specific spatial relationships between parts of the origami model after it is fully folded. The task comprises three types of multiple-choice questions designed to test this ability: 1) **Spatial Pose Localization**: Determining the specific 3D position of a point from the original paper in the final

model, including its pose within a reference frame (e.g., on a table, facing upwards). 2) **Layering Relationship Analysis**: Determining the paper stacking order after folding, requiring analysis of covering relationships during the folding process and identifying how many paper layers form a specific region (e.g., the thickest region). 3) **Geometric Change Analysis**: Predicting how specific geometric features (such as angles, distances, areas, etc.) change from the flat CP diagram to the final folded state. For example, predicting the relative angle or spatial distance between two original line segments after folding. The correct answers for all three question types are obtained using our optimized compiler. Incorrect options are then manually designed. We design 900 multiple-choice questions (300 for each type) for this task. See Appendix B.2 for specific annotation rules.

## 4.4 End-to-End CP Code Generation

This task requires the MLLM to generate corresponding CP code based on a compiled flat layout and an image of the folded shape. Ideally, this CP code should compile into a folded pattern identical to the target shape. To comprehensively evaluate the quality of the generated results, we have designed a multidimensional evaluation framework.

**Compilation Attempt and Evaluation** The CP code generated by the model will first be attempted to be compiled using our origami compiler (see Section 3.2 for details). If the *compilation fails*, the model will return one or more error types. If the *compilation succeeds*, meaning the CP code is syntactically valid, geometrically foldable, and free of self-intersections, and produces a definite folded state, the system will compare the compilation result with the reference result across the following four dimensions:

**1) Topological Structure Similarity (TSS)** This dimension evaluates similarity at the graph theory level by comparing the compiled output. It compares the number of vertices of successfully compiled patterns (score $s_v = e^{-0.5 \frac{|V_{gen} - V_{ref}|}{\min(V_{gen}, V_{ref})}}$), edge connectivity (e.g., similarity of degree distribution, number of connected components), face relationships (e.g., number of faces, distribution of face sizes), and the distribution similarity of crease types ("M", "V", "B", etc.).

**2) Geometric Similarity (GS)** This dimension focuses on the spatial characteristics of the compiled model. It evaluates point position similarity by calculating the bidirectional Hausdorff distance dH between the normalized 3D point sets of the generated and reference compiled models (score $s_p = e^{-k \cdot d_H}$, where k is a sensitivity coefficient, e.g., 5). It assesses angular similarity by comparing the distribution of dihedral angles at the creases, and evaluates size and proportion similarity by comparing the aspect ratios of the overall bounding boxes of the models.

**3) Constraint Satisfaction (CS)** This dimension evaluates whether the successfully compiled CP code, beyond the basic foldability ensured by the compiler, further adheres to the physical and mathematical constraints of origami. This includes comparing the presence and matching degree of critical constraint types (Taco-Taco, Taco-Tortilla, transitivity constraints) and checking for satisfaction of fundamental theorems of local flat-foldability, such as Maekawa's theorem (the difference between the number of mountain creases M and valley creases V around a vertex is $|M - V| = 2$) and Kawasaki's theorem (the sum of the angles $\alpha_i$ of creases around a vertex is $\sum \alpha_i = 2\pi$ or 0).

**4) Final Folded State (FFS)** This dimension directly compares the final 3D model shape compiled from the generated CP with the reference compiled 3D model. It primarily evaluates overall shape similarity by calculating the Hausdorff distance of the point sets, and where possible (if the model provides layering information), compares the layering relationships between facets, including paper stacking order information that may be obtained during the compilation process.

**Total Score:** The final total score $S_{total}$ is a weighted average of the scores $s_{dim}$ from each evaluation dimension: $S_{total} = \sum_{dim} w_{dim} \cdot s_{dim}$. By default, each of the four dimensions accounts for 25% of the weight ($w_{dim} = 0.25$), and $\sum w_{dim} = 1$. This score ranges from 0 to 1 ($S_{total} \in [0, 1]$), reflecting the overall quality of the generated CP code. For more details on the evaluation process, please refer to Appendix D.

| Model | Pattern Prediction | Multi-step Spatial Reasoning | Spatial Relationship Prediction | | |
|---|---|---|---|---|---|
| | | | Spatial Pose Localization | Layering Relationship | Geometric Change |
| *Open-source Models* | | | | | |
| MiniCPM-o 2.6 | $26.99_{\pm0.42}$ | $30.11_{\pm1.54}$ | $28.98_{\pm0.88}$ | $30.50_{\pm1.00}$ | $23.75_{\pm0.09}$ |
| llava-1.5-7b | $27.23_{\pm1.47}$ | $29.05_{\pm1.90}$ | $29.06_{\pm2.71}$ | $30.94_{\pm0.97}$ | $25.51_{\pm0.57}$ |
| deepseek-vl2 | $28.40_{\pm0.07}$ | $30.01_{\pm0.06}$ | $26.71_{\pm1.40}$ | $29.05_{\pm0.23}$ | $24.30_{\pm1.10}$ |
| NVILA-15B | $28.33_{\pm1.09}$ | $32.51_{\pm0.90}$ | $30.60_{\pm1.22}$ | $31.00_{\pm1.53}$ | $26.48_{\pm0.76}$ |
| VideoLLaMA3-7B | $29.01_{\pm1.23}$ | $30.86_{\pm0.14}$ | $29.06_{\pm0.02}$ | $28.74_{\pm1.04}$ | $27.80_{\pm0.35}$ |
| Qwen2.5-VL-7B | $28.40_{\pm0.82}$ | $31.51_{\pm0.30}$ | $28.43_{\pm0.08}$ | $28.05_{\pm0.04}$ | $28.83_{\pm0.72}$ |
| Qwen2.5-VL-32B | $34.15_{\pm0.39}$ | $36.82_{\pm0.48}$ | $33.51_{\pm0.99}$ | $32.59_{\pm0.48}$ | $30.51_{\pm0.15}$ |
| Qwen2.5-VL-72B | $36.29_{\pm0.11}$ | $\mathbf{39.10}_{\pm0.88}$ | $35.68_{\pm1.69}$ | $\mathbf{38.04}_{\pm0.70}$ | $31.89_{\pm0.85}$ |
| InternVL2.5-78B | $\mathbf{36.76}_{\pm0.75}$ | $38.55_{\pm0.08}$ | $\mathbf{38.01}_{\pm0.11}$ | $37.66_{\pm0.13}$ | $\mathbf{32.48}_{\pm0.48}$ |
| *Close-source Models* | | | | | |
| Claude-3.5-Sonnet | $35.89_{\pm1.47}$ | $45.07_{\pm0.64}$ | $39.55_{\pm0.63}$ | $40.19_{\pm0.11}$ | $39.73_{\pm0.10}$ |
| GPT-4o | $42.71_{\pm0.66}$ | $51.81_{\pm0.48}$ | $48.24_{\pm1.73}$ | $\underline{50.42}_{\pm0.59}$ | $46.72_{\pm0.50}$ |
| Gemini2.5-Flash | $35.01_{\pm0.16}$ | $48.92_{\pm0.13}$ | $40.15_{\pm0.60}$ | $39.91_{\pm1.09}$ | $40.01_{\pm1.63}$ |
| Gemini2.5-pro | $\underline{42.68}_{\pm0.14}$ | $\underline{53.45}_{\pm0.74}$ | $\underline{49.06}_{\pm0.07}$ | $47.68_{\pm0.07}$ | $\underline{47.10}_{\pm0.82}$ |
| *Human Performance* | | | | | |
| human(common) | 51.18 | 88.52 | 55.12 | 50.55 | 50.15 |
| human(expert) | 98.45 | 100.00 | 96.44 | 92.10 | 85.38 |

Table 1: Accuracy (%) of various MLLMs on different spatial reasoning tasks. Bold or underlined values indicate best performance across open-source models and all models, respectively.

## 5 Experiments

### 5.1 Models

We evaluate multiple representative MLLMs. For open-source models, we evaluate MiniCPM-o 2.6 [25],NVILA-15B [26], llava-1.5-7b [27], VideoLLaMA3 [28], Qwen2.5-VL-[7B/32B/72B] [29], deepseek-vl2 [30], InternVL2.5-78B [31]. For proprietary models, we evaluate Claude-3.5-Sonnet [32], gpt-4o [33], Gemini2.5-[flash/pro] [34]. For all these models, we adopt the original model and official instruction formats.

### 5.2 Baseline

We recruit two categories of people to complete the first three tasks. The first category consists of five laypersons recruited via a crowdsourcing platform, and the second category comprises three experts with extensive origami experience. Specific details of the human evaluation are provided in Appendix B.3. For the CP code generation task, we adopt the following settings:

**In-context learning** In this setting, we provide the model with detailed task instructions and a set of CP code examples. The instructions will introduce the meaning represented by each part of the CP code and all the constraints that must be followed. MLLMs need to generate the complete CP code in one go based on these instructions and examples.

**Environmental learning** In this setting, MLLMs no longer attempt to generate the complete CP code in one go, but instead engage in iterative interaction with the compiler. Specifically, the MLLM will first perform planning, then generate CP code. The compiler will return its compilation result, and the model then performs inference based on the returned compilation result, subsequently choosing to add or delete creases, iterating in this manner. We set the upper limit of interaction rounds to 10.

**Reinforcement learning** Through a constructed compilation environment, we explore a reinforcement learning approach. We utilize the 471 sets of data mentioned in Section 3.1 for training, sampling data in the same process as in environmental learning. The reward mechanism is set as follows: (1)

| Model | Compilation | | | | | TSS | GS | CS | FFS | Total |
| | CSE | GIF | PSI | AFS | CPR | | | | | |
|---|---|---|---|---|---|---|---|---|---|---|
| *In-context learning* | | | | | | | | | | |
| Qwen2.5-VL-32B | 78.32 | 42.09 | 38.11 | 34.52 | 10.18 | 35.04 | 28.51 | 30.93 | 26.26 | 30.19 |
| Qwen2.5-VL-72B | 80.85 | 44.51 | 40.93 | 37.01 | 14.55 | 37.11 | 31.65 | 33.08 | 28.90 | 32.68 |
| InternVL2.5-78B | 74.12 | 42.17 | 36.01 | 33.91 | 12.84 | 35.04 | 30.67 | 31.95 | 29.04 | 31.68 |
| Claude3.5-Sonnet | 87.36 | 57.94 | 50.12 | 41.62 | 20.73 | 44.02 | 38.99 | 39.21 | 36.87 | 39.77 |
| GPT-4o | 95.03 | 61.13 | 48.28 | 45.25 | 28.56 | 50.06 | 40.57 | 41.58 | 39.06 | 42.82 |
| Gemini2.5-Flash | 83.60 | 50.24 | 46.89 | 40.77 | 18.93 | 42.61 | 40.86 | 37.13 | 36.91 | 39.38 |
| Gemini2.5-pro | 94.47 | 60.06 | 53.41 | 46.01 | 30.03 | 51.51 | 43.71 | 42.68 | 37.28 | 42.80 |
| *Environmental learning* | | | | | | | | | | |
| Qwen2.5-VL-32B | 88.87 | 70.13 | 65.02 | 63.92 | 39.08 | 45.61 | 33.51 | 38.28 | 29.51 | 36.72 |
| Qwen2.5-VL-72B | 90.58 | 77.90 | 68.91 | 66.92 | 43.81 | 48.72 | 38.03 | 40.86 | 34.74 | 40.58 |
| InternVL2.5-78B | 85.23 | 68.92 | 65.81 | 60.01 | 38.34 | 48.91 | 35.37 | 36.72 | 35.02 | 39.00 |
| Claude3.5-Sonnet | 98.05 | 85.89 | 81.74 | 78.52 | 52.90 | 55.82 | 50.21 | 52.15 | 43.66 | 50.46 |
| GPT-4o | **100** | **92.55** | 88.25 | 82.56 | **66.92** | 58.29 | 51.52 | 54.81 | **46.07** | 52.67 |
| Gemini2.5-Flash | 92.51 | 82.81 | 80.03 | 79.93 | 51.94 | 53.01 | 48.86 | 50.95 | 44.91 | 49.43 |
| Gemini2.5-pro | **100** | 90.74 | **92.57** | **84.27** | 65.89 | **60.18** | **52.23** | **56.99** | 45.24 | **53.66** |
| *Reinforcement learning* | | | | | | | | | | |
| Qwen2.5-VL-32B | 91.03 | 72.84 | 70.42 | 68.92 | 45.17 | 49.55 | 39.91 | 42.78 | 38.07 | 42.57 |

Table 2: Results of different MLLMs and methods on the code generation task. Compilation indicates whether compilation is successful, including the probability of no occurrence of the four compilation errors(3.2), as well as the overall compilation pass rate (CPR). When compilation is successful, the similarity in four dimensions(4.4) and the total score are calculated. This score is scaled to [0,100] for ease of presentation.

Intermediate reward: After modifying the code, if compilation is successful, a reward is given based on the quality progress of the current partial CP code ($S_{partial} - S_{partial\_prev}$, where $S_{partial}$ is a quickly evaluated partial quality score), plus a small basic compilation success reward. If compilation fails, a fixed negative penalty is given. (2) Step penalty: A small negative reward is received for each action taken to encourage efficiency. (3) Final reward: After the interaction ends, the result of the evaluation function defined in Section 4.4 serves as the main reward. We adopt TRICO [35] for training on qwen2.5-vl-32B, which is a PPO-based [36], more efficient MLLMs multi-turn reinforcement learning algorithm. Specific training settings and parameters can be found in Appendix E.

## 5.3 Main Results

Tasks 1 to 3 primarily focus on spatial analysis and prediction. The results shown in Table 1 are the average of three runs for different MLLMs, from which we observe that: 1) For MLLMs, ORIGAMISPACE is a challenging task; the performance of poor-performing models is close to random guessing (25%), and even for the best-performing models, there is a significant gap compared to human performance, especially in multi-step spatial reasoning. 2) Despite the different task types, the relative performance ranking of various models largely remains consistent, with Gemini 2.5-pro and GPT-4o demonstrating the best spatial reasoning ability. 3) Human experts perform well on all tasks, demonstrating the task's upper bound. 4) MLLMs perform worst on the Spatial Relationship Prediction task, especially the sub-tasks involving Geometric Change, indicating significant difficulty for models in understanding fine-grained, internal spatial structures.

Table 2 presents the results of different methods and models on Task 4. We observe the following: 1) Impact of learning settings: The results clearly indicate the significant impact of learning settings on performance. In-context learning shows relatively limited performance. Environmental learning brings significant performance improvements, demonstrating that through iterative interaction with the compiler, planning, and trial-and-error based on feedback, models can overcome the limitations

of one-shot generation. Reinforcement learning shows potential, as the trained Qwen2.5-VL-32B surpassed the performance of a 72B model. 2) There are significant performance differences among different models, with top-tier closed-source models exhibiting the best spatial reasoning capabilities.

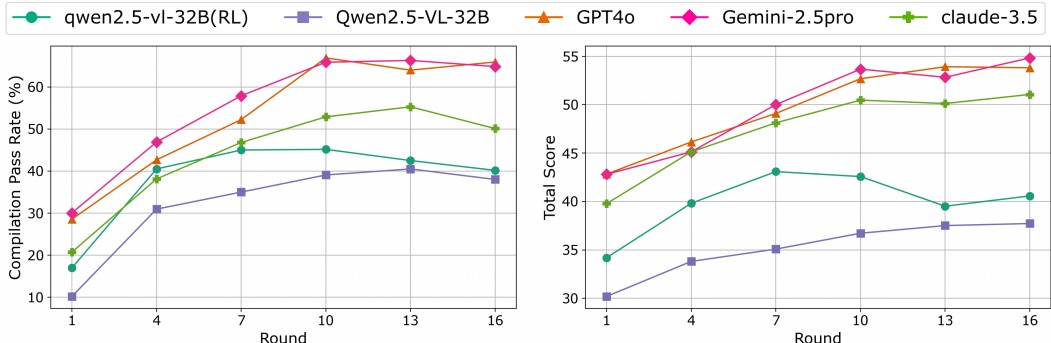

Figure 3: The impact of interaction rounds on the compilation pass rate and total score of different models.

## 5.4  Impact of Mathematical Constraints

Mathematical constraints present a primary challenge in generating valid CP codes for the ORIGAMIS-PACE task. Table 2 indicates that failing to satisfy constraints is the main bottleneck for compilation failures; even when provided with detailed instructions, models struggle to strictly adhere to these complex rules, leading to persistently high compilation failure rates. Interactive processes with the environment enhance models' ability to follow constraints, demonstrating that models can learn and internalize rules from feedback. Compared to environmental learning, reinforcement learning also shows improvement in constraint satisfaction, proving the effectiveness of specific reward mechanisms. However, even with interactive learning, precisely satisfying all mathematical constraints remains a significant challenge for top-tier models (such as GPT-4o and Gemini 2.5-pro, whose *constraint satisfaction* score is only 56.99% under environmental learning settings). This reveals MLLMs' deficiencies in deep multi-step geometric and layering reasoning and highlights the value of the fine-grained feedback and constraint satisfaction evaluation introduced in this study.

## 5.5  Impact of Interaction Rounds in Environmental Learning

Figure 3 illustrates the impact of interaction rounds on model performance across different dimensions under the environmental learning setting. We observe that as the number of interaction rounds increases, model performance improves in various aspects, particularly the compilation pass rate. However, performance tends to saturate after 8-10 rounds, indicating that interaction primarily helps overcome initial learning obstacles but struggles to break through the model's inherent bottlenecks. Weaker models, limited by their understanding capabilities, reach their upper limit in fewer rounds. The reinforcement learning-trained Qwen2.5-VL-32B also follows a similar trend, but due to policy optimization, it may reach its performance ceiling in fewer rounds.

## 6  Conclusion

In this paper, we introduce ORIGAMISPACE, a novel benchmark specifically designed to address the underexplored areas of multi-step spatial reasoning and constraint adherence in Multimodal Large Language Models (MLLMs). Leveraging the inherent complexities of origami, ORIGAMISPACE provides 350 meticulously curated data instances and an enhanced compilation program to facilitate in-depth evaluation. The benchmark features four challenging tasks, including pattern prediction, spatial relationship prediction, multi-step spatial reasoning, and end-to-end code generation, making it the first to assess MLLMs' multi-step spatial reasoning under rigorous mathematical constraints. Our comprehensive evaluation of existing MLLMs and exploration of reinforcement learning methods for code generation highlight the utility of ORIGAMISPACE in not only assessing current capabilities but also in paving new ways to enhance the spatial intelligence of MLLMs.

## Acknowledgments and Disclosure of Funding

We wish to express our sincere gratitude to several individuals and communities for their invaluable contributions to this research. We are deeply grateful for the enthusiastic support and engagement from the global origami community, particularly the many knowledgeable members active on Discord servers dedicated to origami and Baidu Tieba's origami forums. Their discussions and shared resources were highly beneficial. Special thanks are extended to Niels Stoermer, founder of the Origami Database, for his generosity in providing foundational data and for his instrumental assistance in connecting us with a wider network of origami experts. We are also immensely thankful to QingLiang and Jason Ku for offering their expert advice and constructive feedback on the development and refinement of our origami compiler; their insights were crucial.

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

# A  Dataset

The ORIGAMISPACE comprises a total of 350 data entries, covering various types of origami. We have categorized these based on the required number of folding steps into Easy (3-9 steps), Medium (10-19 steps), and Hard (20-30 steps). Tables 3, 4, and 5 respectively display all the data for these three difficulty levels, including the origami design name, its category, and the number of folding steps required. All our data are public data or authorized by the original websites and data sources, with no potential infringement risks.

Table 3: Easy Origami Models (3-9 Steps)

| Easy Origami Models (3-9 Steps) | |
| --- | --- |
| 1. Triangle - Geometry - 3 | 2. Square base fold - Geometry - 4 |
| 3. Mountain - Nature - 4 | 4. Letter I - Alphabet - 3 |
| 5. Number 1 - Numbers - 3 | 6. Minus Sign - Symbols - 3 |
| 7. Bird Beak - Animals - 4 | 8. Letter L - Alphabet - 4 |
| 9. Number 7 - Numbers - 4 | 10. Cross Mark/X - Symbols - 4 |
| 11. Plus Sign - Symbols - 4 | 12. Diamond shape - Geometry - 5 |
| 13. Water Drop - Nature - 5 | 14. Trapezoid - Geometry - 5 |
| 15. Lucky Star strip prep - Decorations - 5 | 16. Comma symbol - Symbols - 5 |
| 17. Single French Fry - Food - 5 | 18. Letter C - Alphabet - 5 |
| 19. Fish Fin - Animals - 5 | 20. Check Mark - Symbols - 5 |
| 21. Nail - Tools - 5 | 22. Simple Envelope - Items - 6 |
| 23. Small Flag - Decorations - 6 | 24. Simple Leaf - Plants - 6 |
| 25. Arrow - Symbols - 6 | 26. Band-aid - Items - 6 |
| 27. Screw - Tools - 6 | 28. Letter F - Alphabet - 6 |
| 29. Number 2 - Numbers - 6 | 30. Number 4 - Numbers - 6 |
| 31. Bread Slice - Food - 6 | 32. Plate - Items - 6 |
| 33. Simple Cloud - Nature - 6 | 34. Simple Ring band - Accessories - 6 |
| 35. Ice Lolly/Popsicle Stick - Food - 6 | 36. Simple Coaster - Items - 7 |
| 37. Pointed Bookmark - Items - 7 | 38. Paper Dart - Toys - 7 |
| 39. Simple Heart - Decorations - 7 | 40. Fox - Animals - 7 |
| 41. Iceberg - Nature - 7 | 42. Bone - Items - 7 |
| 43. Simple Pen/Pencil outline - Items - 7 | 44. Simple Screwdriver outline - Tools - 7 |
| 45. Letter E - Alphabet - 7 | 46. Number 3 - Numbers - 7 |
| 47. Letter Z - Alphabet - 7 | 48. Simple Fish - Animals - 7 |
| 49. Simple Mushroom - Plants - 7 | 50. Simple Radish/Carrot top - Plants - 7 |
| 51. House Outline - Items - 7 | 52. Simple Tent/Teepee - Items - 7 |
| 53. Ice Cream Cone base - Food - 7 | 54. Pointy Hat - Clothing - 7 |
| 55. Crescent Moon - Nature - 7 | 56. Candle - simple - Items - 7 |
| 57. Simple Ghost - Decorations - 7 | 58. Boomerang - simple V - Toys - 7 |
| 59. Cheese Slice - Food - 7 | 60. Simple Shovel outline - Tools - 7 |
| 61. Simple Cup - Items - 8 | 62. Simple Boat - Items - 8 |
| 63. Simple Dog Face - Animals - 8 | 64. Simple Cat Face - Animals - 8 |
| 65. Simple Pig Face - Animals - 8 | 66. Traditional Cup/Masu Box base - Traditional - 8 |
| 67. Apple Core shape - Food - 8 | 68. Simple Necktie - Clothing - 8 |
| 69. Dinosaur Footprint - Animals - 8 | 70. Clover/Shamrock - Plants - 8 |
| 71. Simple Butterfly - Animals - 8 | 72. Computer Mouse - simple - Items - 8 |
| 73. Simple Crown band - Clothing - 8 | 74. Letter A - Alphabet - 8 |
| 75. Number 0 - Numbers - 8 | 76. Frisbee - flat circle - Toys - 8 |
| 77. Croissant shape - very simple - Food - 8 | 78. Egg shape - flat - Food - 8 |
| 79. Sandwich - triangle cut - Food - 8 | 80. Onigiri/Rice Ball shape - Food - 8 |
| 81. Lollipop - circle on stick - Food - 8 | 82. Simple Hammer outline - Tools - 8 |
| 83. Simple Saw outline - Tools - 8 | 84. Tadpole - Animals - 8 |
| 85. Simple Bow - Decorations - 8 | 86. Simple Pinwheel base - Toys - 9 |
| 87. Simple Book - Items - 9 | 88. Snail Shell - Animals - 9 |
| 89. Simple Snake - Animals - 9 | 90. Tulip Head - Plants - 9 |
| 91. Simple Shield - Toys - 9 | 92. Bird Silhouette - very simple - Animals - 9 |
| 93. Square Coaster - Items - 9 | 94. Flat Christmas Tree - Plants - 9 |
| 95. Number 8 - Numbers - 9 | 96. Fishbone - Animals - 9 |
| 97. Bamboo Shoot - Plants - 9 | 98. Lemon slice - Food - 9 |
| 99. Donut - flat with hole - Food - 9 | 100. Pretzel shape - very simple - Food - 9 |
| 101. Fried Egg - flat - Food - 9 | 102. Hot Dog in bun - flat - Food - 9 |
| 103. Sushi Roll - simple cylinder end - Food - 9 | 104. Tea Bag with string - Food - 9 |
| 105. Simple Vase outline - Items - 9 | 106. Simple Wrench outline - Tools - 9 |
| 107. Simple Axe outline - Tools - 9 | 108. Dinosaur Egg - Animals - 9 |
| 109. Bow Tie - Clothing - 9 | 110. Candy Cane - Food - 9 |
| 111. Letter J - Alphabet - 9 | 112. Stop Sign - octagon shape - Symbols - 9 |

Table 4: Medium Difficulty Origami Models (10-19 Steps)

| Medium Difficulty Origami Models (10-19 Steps) | |
|---|---|
| 1. Dog body - Animals - 10 | 2. Pig body - Animals - 10 |
| 3. Swan - Animals - 10 | 4. Goldfish - Animals - 10 |
| 5. Butterfly - common - Animals - 10 | 6. Starfish - Animals - 10 |
| 7. Sun with rays - Nature - 10 | 8. House with roof - Items - 10 |
| 9. Photo Frame - Items - 10 | 10. Sword - Toys - 10 |
| 11. Sailboat - Items - 10 | 12. Classic Glider - Toys - 10 |
| 13. Triangular Box base - Items - 10 | 14. Shuriken - single piece - Traditional - 10 |
| 15. Kimono - flat - Traditional - 10 | 16. Strawberry - Food - 10 |
| 17. Watermelon Slice - Food - 10 | 18. Banana - Food - 10 |
| 19. Shirt - Clothing - 10 | 20. Simple Tree - flat - Plants - 10 |
| 21. Acorn - Plants - 10 | 22. Witch Hat - Clothing - 10 |
| 23. Sock/Stocking - Clothing - 10 | 24. Ring with simple gem - Accessories - 10 |
| 25. Letter B - Alphabet - 10 | 26. Modular Box Corner Unit - simple - Modular - 10 |
| 27. Easter Egg Stand - Decorations - 10 | 28. Thermometer - simple - Items - 10 |
| 29. Letter H - Alphabet - 10 | 30. Number 6 - Numbers - 10 |
| 31. Number 9 - Numbers - 10 | 32. Caterpillar - simple segments - Animals - 10 |
| 33. Mitten - Clothing - 10 | 34. Letter K - Alphabet - 10 |
| 35. Simple Sofa - front view - Furniture - 10 | 36. Pigeon - Animals - 11 |
| 37. Duck - Animals - 11 | 38. Pinwheel - functional - Toys - 11 |
| 39. Pouch - simple - Items - 11 | 40. Dress - simple - Clothing - 11 |
| 41. Pear - Food - 11 | 42. Seagull - simple flying - Animals - 11 |
| 43. Slipper - flat - Clothing - 11 | 44. Mobile Phone - flat - Items - 11 |
| 45. Popsicle - Food - 11 | 46. Number 5 - Numbers - 11 |
| 47. Signpost - Items - 11 | 48. Scarf - Clothing - 11 |
| 49. Simple Bed - top view - Furniture - 11 | 50. Cat body - Animals - 12 |
| 51. Angelfish - Animals - 12 | 52. Ladybug - Animals - 12 |
| 53. Crane - traditional - Animals - 12 | 54. Bat - Animals - 12 |
| 55. Chair - Furniture - 12 | 56. Lantern - simple flat - Items - 12 |
| 57. Airplane - dart style - Toys - 12 | 58. Masu Box - Traditional - 12 |
| 59. Wallet/Coin Purse - simple - Items - 12 | 60. Samurai Helmet/Kabuto - Traditional - 12 |
| 61. Apple shape - Food - 12 | 62. Pants - Clothing - 12 |
| 63. Cupcake paper - Food - 12 | 64. Four-Leaf Clover - Plants - 12 |
| 65. Lily flower - simple - Plants - 12 | 66. Cactus - simple - Plants - 12 |
| 67. Wheat stalk - simple - Plants - 12 | 68. Glasses - Accessories - 12 |
| 69. Gingerbread Man - flat - Food - 12 | 70. Geometric Pattern tile - Geometry - 12 |
| 71. Cake Slice - flat - Food - 12 | 72. Fish Bowl - flat simple - Items - 12 |
| 73. Letter G - Alphabet - 12 | 74. Pirate Hat - simple flat - Clothing - 12 |
| 75. Letter M - Alphabet - 12 | 76. Simple Street Lamp post - Items - 12 |
| 77. Jumping Frog base - Animals - 13 | 78. Fan - Items - 13 |
| 79. Ice Cream with scoop - Food - 13 | 80. Seal - Animals - 13 |
| 81. Monkey Face - Animals - 13 | 82. Koala Face - Animals - 13 |
| 83. Cherries - pair - Food - 13 | 84. Key - Items - 13 |
| 85. Tent - A-frame - Items - 13 | 86. Medal - Decorations - 13 |
| 87. Easter Bunny Face - Decorations - 13 | 88. Coffee Mug - Items - 13 |
| 89. Traffic Light - simple - Items - 13 | 90. Chef Hat - simple flat - Clothing - 13 |
| 91. Flying Saucer - simple - Toys - 13 | 92. Bear Face - Animals - 14 |
| 93. Maple Leaf - Plants - 14 | 94. Chicken - simple - Animals - 14 |
| 95. Grapes - simple bunch - Food - 14 | 96. Tulip with stem - Plants - 14 |
| 97. Crown - fuller - Clothing - 14 | 98. Gift Box - flat with bow - Decorations - 14 |
| 99. Baseball Cap - flat - Clothing - 14 | 100. Computer Monitor - flat - Items - 14 |
| 101. Finger Puppet Bear - Toys - 14 | 102. Simple Tree Ornament - Decorations - 14 |
| 103. Hamburger - simple layers - Food - 14 | 104. Dog House - simple front - Items - 14 |
| 105. Mailbox - simple - Items - 14 | 106. Top Hat - simple - Clothing - 14 |
| 107. Rabbit - Animals - 15 | 108. Penguin - Animals - 15 |
| 109. Snake - Coiled Snake - Animals - 15 | 110. Lion Face - Animals - 15 |
| 111. Tiger Face - Animals - 15 | 112. Table - simple - Furniture - 15 |
| 113. Rocket - simple - Toys - 15 | 114. Pumpkin - flat - Food - 15 |
| 115. Rose - easy flat - Plants - 15 | 116. Woodpecker - simple head - Animals - 15 |
| 117. Daisy - simple - Plants - 15 | 118. Boot - simple - Clothing - 15 |
| 119. Diamond shape - faceted look - Decorations - 15 | 120. Modular Star - 3 simple points - Decorations - 15 |
| 121. Halloween Bat - hanging - Decorations - 15 | 122. Pen Holder - very simple cylinder - Items - 15 |
| 123. Bird House - simple front - Items - 15 | 124. Ladies Hat - wide brim simple - Clothing - 15 |
| 125. Dragonfly - simple - Animals - 16 | 126. Sunflower - simple face - Plants - 16 |
| 127. Pineapple - simple - Food - 16 | 128. Winged Heart - Decorations - 16 |
| 129. Lock - simple - Items - 16 | 130. Woven Mat - small 2x2 strip - Geometry - 16 |
| 131. Teapot - simple flat - Items - 16 | 132. Compass Rose - 4 points - Symbols - 16 |
| 133. Bench - Furniture - 16 | 134. Guitar - simple flat - Musical Instruments - 17 |
| 135. Carnation - simplified - Plants - 17 | 136. Snowflake - simple 6-point - Decorations - 17 |
| 137. Bucket/Pail - Items - 17 | 138. Firefighter Helmet - simple - Clothing - 17 |
| 139. Cicada - simple - Animals - 18 | 140. Squirrel - simple - Animals - 18 |
| 141. Lotus Flower - simple - Plants - 18 | 142. Car - side view, simple - Vehicles - 18 |
| 143. Piano - simple upright - Musical Instruments - 18 | 144. Poinsettia - simple layer - Plants - 18 |
| 145. 3D Star - simple module - Decorations - 18 | 146. Tissue Box Cover - simple sleeve - Items - 18 |

Table 5: Hard Origami Models (20-30 Steps)

| Difficult Origami Models (20-30 Steps) | |
|---|---|
| 1. Elephant - standing - Animals - 20 | 2. Bookend - simple L-shape, thick - Items - 20 |
| 3. Drum - simple - Musical Instruments - 20 | 4. Torch with flame - Items - 20 |
| 5. Giraffe - standing - Animals - 22 | 6. Hedgehog - Animals - 22 |
| 7. Lidded Box - separate lid & base, simple - Items - 22 | 8. Spaceship - simple rocket style - Vehicles - 22 |
| 9. Binoculars - Items - 22 | 10. Phone Stand - functional - Items - 22 |
| 11. Pyramid - more detailed base - Architecture - 22 | 12. Scroll - open - Items - 22 |
| 13. Bear - standing - Animals - 23 | 14. Shrimp/Prawn - Animals - 23 |
| 15. Lighthouse - Architecture - 23 | 16. Sailboat - more detailed - Vehicles - 23 |
| 17. Hourglass shape - Items - 23 | 18. Dog Toy - squeaky bone shape - Toys - 23 |
| 19. Floor Lamp - Furniture - 23 | 20. Camel - Animals - 24 |
| 21. Crab - Animals - 24 | 22. Hot Air Balloon - simple 3D - Vehicles - 24 |
| 23. Camera - simple 3D body - Items - 24 | 24. Trophy Cup - Items - 24 |
| 25. Bridge - simple arch - Architecture - 24 | 26. Vase - with some shaping - Items - 24 |
| 27. Shoji Screen - simple panel - Traditional - 24 | 28. Horse - standing - Animals - 25 |
| 29. Hippopotamus - Animals - 25 | 30. Shark - Animals - 25 |
| 31. Bee - detailed wings - Animals - 25 | 32. Owl - with features - Animals - 25 |
| 33. Treasure Chest - simple - Items - 25 | 34. Church - simple front - Architecture - 25 |
| 35. Robot - boxy - Toys - 25 | 36. Microphone with stand base - Items - 25 |
| 37. Tower/Rook chess piece shape - Toys - 25 | 38. Kettle - Items - 25 |
| 39. Photo Frame - standing type - Items - 25 | 40. Sofa - more detailed - Furniture - 25 |
| 41. Panda - sitting - Animals - 26 | 42. Kangaroo with joey pouch outline - Animals - 26 |
| 43. Seahorse - Animals - 26 | 44. Flamingo - Animals - 26 |
| 45. Truck - simple 3D profile - Vehicles - 26 | 46. Eiffel Tower - simplified flat - Landmarks - 26 |
| 47. Harp - simplified profile - Musical Instruments - 26 | 48. Tent - more complex dome like - Items - 26 |
| 49. Snowman - Decorations - 26 | 50. Wolf - howling pose - Animals - 27 |
| 51. Turtle - with shell detail - Animals - 27 | 52. Eagle - spread wings - Animals - 27 |
| 53. Windmill building with vanes - Architecture - 27 | 54. Violin - simplified profile - Musical Instruments - 27 |
| 55. Backpack - with straps - Items - 27 | 56. Dragonfly - more detailed - Animals - 27 |
| 57. Sports Car - simple profile - Vehicles - 27 | 58. Potted Plant - simple - Plants - 27 |
| 59. Lion - standing - Animals - 28 | 60. Deer/Stag - Animals - 28 |
| 61. Crocodile/Alligator - simple form - Animals - 28 | 62. Spider - 8 legs - Animals - 28 |
| 63. Parrot - on perch - Animals - 28 | 64. Pentagonal Box - simple - Items - 28 |
| 65. Train Engine - simple profile - Vehicles - 28 | 66. Castle - simple front - Architecture - 28 |
| 67. Old Telephone - receiver and body - Items - 28 | 68. Saxophone - simplified profile - Musical Instruments - 28 |
| 69. Accordion - simplified - Musical Instruments - 28 | 70. Butterfly - more detailed - Animals - 28 |
| 71. Christmas Wreath - simple modular - Decorations - 28 | 72. Unicorn - simple standing - Animals - 28 |
| 73. Laptop - open - Items - 28 | 74. Rhinoceros - Animals - 29 |
| 75. Peacock - simplified tail - Animals - 29 | 76. Pterodactyl - simple - Animals - 29 |
| 77. Fire Truck - basic shape - Vehicles - 29 | 78. Police Car - basic shape - Vehicles - 29 |
| 79. Ambulance - basic shape - Vehicles - 29 | 80. Grand Piano - simplified top view - Musical Instruments - 29 |
| 81. Clownfish - Animals - 29 | 82. Ice Cream Truck - simple profile - Vehicles - 29 |
| 83. Lotus - multi-petal - Plants - 29 | 84. Octopus - with 8 simple tentacles - Animals - 30 |
| 85. Scorpion - Animals - 30 | 86. Dinosaur T-Rex - simple standing - Animals - 30 |
| 87. Hexagonal Box - simple - Items - 30 | 88. Bicycle - very simplified profile - Vehicles - 30 |
| 89. Motorcycle - very simplified profile - Vehicles - 30 | 90. Pirate Ship - simplified - Vehicles - 30 |
| 91. Double Decker Bus - simple profile - Vehicles - 30 | 92. Reindeer - simple standing - Animals - 30 |

# B  Manual annotation

## B.1  Annotation Rules for Pattern Prediction Task

The primary goal of this annotation task is to create challenging yet fair incorrect options for multiple-choice questions (MCQs). For each given Crease Pattern (CP) diagram and its known correct folded 3D shape, annotators are required to design three distinct incorrect shape options. These options, along with the correct one, will form an MCQ designed to evaluate a model's ability to predict the 3D shape from the CP. The following rules must be strictly adhered to when designing these incorrect options:

### B.1.1  Rule 1: Ensure Visual Distinguishability

Each incorrect option must be easily and clearly distinguishable visually from the correct folded shape. The purpose is to prevent ambiguity where an incorrect option might be confused with the correct one due to only subtle visual differences.

**Guideline:**

- The overall silhouette, major components, and general form of the incorrect option should be significantly different from those of the correct option.
- Avoid creating incorrect options that are merely slight modifications, re-orientations, or minor proportional changes of the correct shape.

**Example:**

- If the correct shape is an *origami crane*:
  - An incorrect option that is another bird in a very similar pose (e.g., a crane with wings slightly more elevated versus wings fully spread, if the overall form remains highly similar) might be **unsuitable** if it's not clearly visually distinct at a glance.
  - A **suitable** incorrect option would be an *origami box*, an *origami boat*, or an *origami star*, as these are visually very different from a crane.

### B.1.2   Rule 2: Maintain Conceptual Distinctness

Incorrect options should not be variations of the same concept or fall within the same narrow semantic category as the correct option. They should represent fundamentally different objects or ideas. This rule ensures the task tests the prediction of the specific shape, not fine-grained classification within a single conceptual group.

**Guideline:**

- If the correct option is a specific type of animal, incorrect options should not be other animals that are closely related (e.g., from the same family) or share very similar overarching characteristics.
- Strive for incorrect options that belong to different conceptual categories than the correct option (e.g., animal vs. inanimate object vs. geometric form).

**Example:**

- If the correct shape is an *origami cat*:
  - Incorrect options such as *Lion*, *Tiger*, or *Leopard* are **unsuitable** because they are all felines and thus variations of the same core concept ("large cat" or "wild cat" as opposed to "domestic cat").
  - **Suitable** incorrect options could be an *origami airplane*, an *origami hat*, or an *origami fish* (assuming the 'fish' is a distinctly different concept from 'cat' within the context of common origami figures).

### B.1.3   Rule 3: Ensure Crease Pattern Plausibility

While incorrect, the alternative shapes should be plausible outcomes that could potentially be folded from a Crease Pattern that bears some relationship to the given CP diagram. This means an incorrect option might be a shape that could result from misinterpreting some creases, omitting a few key folds, or simplifying the original pattern. The objective is to create distractors that are not arbitrary but reflect potential, albeit erroneous, folding paths from a CP similar to the one provided.

**Guideline:**

- Consider what alternative, simpler, or related shapes might emerge if certain folds in the CP are ignored, if mountain and valley folds are confused, or if a common base derived from the CP is completed into a different known figure.
- The incorrect option's implied CP should not be drastically more complex or entirely unrelated to the structural elements suggested by the given CP. It should ideally represent a shape that an intermediate folder might erroneously produce when attempting the correct model or a related one.

**Example:**

- Given a CP diagram for a relatively simple *origami boat*:

– A **suitable** incorrect option could be an *origami hat* (e.g., a traditional paper hat like a "samurai helmet" or a simple party hat). Many simple hats share foundational folds or bases (like the water bomb base or a preliminary fold variation) with simple boats, or their CPs can be derived by altering or omitting a few creases from a boat's CP.

– An **unsuitable** incorrect option might be a highly complex *origami insect* or a multi-piece *modular origami ball* if the provided CP is for a simple, single-sheet boat. The CP for such complex figures would likely be vastly different and far more intricate, making them implausible alternatives based on the given simple CP.

**Summary for Annotators Creating Incorrect Options:** For each CP diagram and its corresponding correct folded shape, you are to design three unique incorrect shape options. Before finalizing these options, please verify each one against the following three criteria:

1. **Visual Distinguishability:** Is the incorrect option clearly visually different from the correct shape?

2. **Conceptual Distinctness:** Is the incorrect option conceptually different from the correct shape, avoiding mere variations of the same theme?

3. **Crease Pattern Plausibility:** Is the incorrect option a shape that could plausibly (even if incorrectly) be derived from the provided CP or a closely related CP (e.g., through simplification or common error)?

Adherence to these rules is crucial for creating high-quality and effective multiple-choice questions for the Pattern Prediction evaluation task.

## B.2 Annotation Guidelines for Incorrect Option Generation in Spatial Relationship Prediction Task

This section outlines the rules for annotators tasked with designing incorrect options for the Spatial Relationship Prediction task. For each Crease Pattern (CP) diagram, questions are posed about the spatial properties of the final folded origami model. While correct answers are generated by an optimized compiler, annotators must manually create three plausible yet incorrect options for each question to form a multiple-choice question (MCQ). The aim is to generate distractors that effectively test a model's nuanced understanding of 3D spatial relationships post-folding.

The task comprises three types of questions. Below are specific guidelines for designing incorrect options for each type:

### B.2.1 Type 1: Spatial Pose Localization

This question type requires predicting the specific 3D position and/or pose (orientation) of a designated point (or feature) from the original flat paper once the model is fully folded. The pose might be described relative to a global reference frame (e.g., on a table, with a specific part facing upwards).

**Guidelines for Designing Incorrect Options:**

- **Plausible Positional Errors:**
  - Offer coordinates that are slightly offset from the correct 3D position (e.g., incorrect by a small delta in one or more axes, located in an adjacent quadrant, or on a wrong but nearby surface).
  - Suggest a position that would be correct if a key fold were made inaccurately (e.g., a mountain fold treated as a valley, an incorrect fold angle, or slight misalignment of layers).
  - Propose the final position of a different, perhaps nearby or symmetrically opposite, salient point from the original CP.

- **Plausible Pose Errors (if orientation is part of the question):**
  - Provide options with the correct 3D position but an incorrect orientation (e.g., correct $(x, y, z)$ coordinates, but the point/surface faces downwards instead of upwards, or is rotated $90°$ incorrectly).

– Offer an orientation that is a common simplification (e.g., aligned perfectly with a major axis when it's actually slightly tilted).

- **Symmetry-based Errors:** For CPs/models exhibiting symmetry, an incorrect option could be the symmetrical counterpart of the correct position or pose.

- **Reference Frame Confusion:** Offer a position or pose that is correct relative to a local part of the origami model but incorrect within the specified global reference frame, or vice-versa.

**Example:** Suppose a specific vertex 'P' on the CP is queried for its final 3D coordinates $(x, y, z)$ and the direction its local paper surface is facing (e.g., 'upwards'), relative to a table it sits on. The correct answer (from compiler) is $(10, 5, 3)$, local surface facing 'upwards'.

- **Suitable Incorrect Options could be:**
    - $(10, 5, 0)$, local surface facing 'upwards' (Incorrect Z-coordinate, perhaps implying it's on the table surface when it's elevated).
    - $(10, 5, 3)$, local surface facing 'downwards' (Correct position, but incorrect orientation).
    - $(-10, 5, 3)$, local surface facing 'upwards' (A symmetrical position if the model has YZ plane symmetry and origin is centered).
    - The final coordinates and pose of an adjacent vertex 'Q' from the CP.

- **Unsuitable Incorrect Options:** Random coordinates or orientations with no plausible relation to the model's scale, structure, or folding process.

### B.2.2 Type 2: Layering Relationship Analysis

This question type focuses on the internal structure of the folded model, specifically the stacking order of paper layers or the number of layers at a particular region (e.g., identifying the thickest region or counting layers at a specific point).

**Guidelines for Designing Incorrect Options:**

- **For Number of Layers Questions:**
    - Offer layer counts that are slightly off from the correct number (e.g., correct count $\pm 1$ or $\pm 2$ layers).
    - Propose the layer count of an adjacent or visually similar region in the folded model.
    - Suggest a count that might result from overlooking some hidden internal layers or, conversely, double-counting some visible folded edges as separate layers.
    - If the question asks to identify the "thickest region" from a set of options, incorrect options should be other regions that are also thick, but not maximally so, or regions that appear thick but are not.

- **For Stacking Order Questions:**
    - Provide plausible but incorrect permutations of the layer sequence. For example, if the correct top-to-bottom order of layers (referenced by their original CP surface labels like S1, S2, S3) is S1-S3-S2, an incorrect option could be S1-S2-S3 or S2-S1-S3.
    - Suggest an order that would result if a specific flap were tucked differently during folding (e.g., a flap going over another flap instead of under it).
    - Offer an incomplete order (e.g., missing one or more layers from the sequence in that region) or an order that incorrectly includes layers not present in that specific stack.

**Example:** Question: "How many layers of paper form the central part of the crane's body?" Correct answer (from compiler): 8 layers.

- **Suitable Incorrect Options could be:**
    - 6 layers (Plausible underestimation, perhaps missing some internal folds).
    - 7 layers (Close, but incorrect).
    - 10 layers (Plausible overestimation, perhaps counting edges).
    - 4 layers (Number of layers in the crane's wing, a different region).

Question: "Consider a point X on the wing of a folded paper airplane. Starting from the externally visible top surface at X, what is the order of the original paper surfaces (labeled S1, S2, S3, S4 on the CP) one would pass through if drilling perpendicularly downwards through all layers at X?" Correct answer (from compiler): S1, S4, S2.

- **Suitable Incorrect Options could be:**
    - S1, S2, S4 (A common misremembered or simplified stacking).
    - S4, S1, S2 (Incorrect starting layer or internal order).
    - S1, S4 (Incomplete, missing the bottom layer S2).

### B.2.3   Type 3: Geometric Change Analysis

This question type involves predicting how specific geometric features (e.g., angles between lines, distances between points, areas of surfaces) change from their state in the flat CP diagram to their state in the final 3D folded model.

**Guidelines for Designing Incorrect Options:**

- **Value from Original CP:** A very common and effective incorrect option is to offer the original geometric value as it was on the flat CP diagram (e.g., if an angle is $90°$ on the CP but becomes $45°$ in 3D, then $90°$ is a strong distractor). This tests whether the model understands that geometric properties transform during folding.

- **Plausible Estimations or Miscalculations:**
    - For angles: Provide common angles (e.g., $30°, 45°, 60°, 90°, 180°$) that might appear correct upon a cursory visual inspection of the folded form, or angles that result from assuming a simplified 3D configuration (e.g., assuming perpendicularity or parallelism where it doesn't exactly exist).
    - For distances: Offer distances measured along the paper surface instead of the true Euclidean distance through 3D space (or vice-versa, depending on the question's phrasing). Suggest distances that might result from slight errors in visualizing the 3D form, such as ignoring foreshortening or using dimensions from a 2D projection.
    - For areas: Propose areas that don't account for overlaps of paper in the folded state, or the area of a 2D projection rather than the true 3D surface area (if the latter is specified). An area that results from a miscalculation of how a shape transforms (e.g., halving an area when it should be less or more).

- **Qualitative Change Errors:** If the question is about the nature of change (e.g., "Does distance X increase, decrease, or stay the same?"), incorrect options could be the opposite type of change, or "stays the same" when there is indeed a significant change.

- **Values from Unrelated or Different Parts:** Offer a geometric value (angle, distance, area) that is correct for a different feature or part of the folded model, or for a different but related origami model.

**Example:** Question: "Two line segments L1 and L2 are parallel on the CP diagram and are 5 cm apart. In the final folded model, these segments become two adjacent edges of a wing. What is the approximate angle between the segments L1 and L2 in the folded state?" Correct answer (from compiler): $60°$.

- **Suitable Incorrect Options could be:**
    - $0°$ (Implying they remain parallel, i.e., no change from CP state regarding their relative orientation).
    - $90°$ (A common angle in man-made objects and some origami steps, could be a plausible guess).
    - $45°$ (Another common angle, plausible visual estimate).

Question: "A defined square region on the CP has an area of 16 cm$^2$. After folding, this region forms part of a curved surface. What is the approximate surface area of this region in the 3D model?" Correct answer (from compiler): 16 cm$^2$ (assuming no stretching/shrinking of paper, the intrinsic surface area remains the same, though its projected area might change).

- **Suitable Incorrect Options could be:**
    - $8 \text{ cm}^2$ (Perhaps confusing with a projected area that is halved).
    - $12 \text{ cm}^2$ (A value less than original, implying shrinkage or significant overlap not intrinsic to the region itself).
    - $20 \text{ cm}^2$ (A value more than original, implausible without stretching). * (Note: If the question was about *projected area*, then $16 \text{ cm}^2$ could be an incorrect option if the projection foreshortens it).

**General Summary for Annotators Designing Incorrect Options:** For each question across these three types, remember the following overarching principles when designing your three incorrect options:

1. **Understand the Query:** First, be absolutely clear about what the question is asking regarding the folded CP and what the compiler-generated correct answer is.
2. **Plausibility is Key:** Incorrect options should appear as reasonable possibilities to someone who might have a slight misunderstanding of the folding process, 3D geometry, or spatial reasoning. Avoid options that are trivially wrong, absurd, or completely random.
3. **Ensure Clear Incorrectness:** While plausible, each incorrect option must be demonstrably wrong upon careful analysis based on the correct folding sequence and 3D geometry.
4. **Introduce Variety in Errors:** The set of three incorrect options should ideally probe different potential misunderstandings or types of errors (e.g., one based on CP value, one on slight miscalculation, one on conceptual error).
5. **Maintain Consistency:** Ensure that the format of your incorrect options (e.g., units, precision of numbers, terminology) is consistent with the format of the correct answer.

By following these guidelines, you will help create high-quality multiple-choice questions that rigorously and fairly evaluate a model's capabilities in spatial relationship prediction for origami.

### B.3 Human evaluation

For the manual evaluation of the first three tasks, we recruited evaluators from two different categories. The first category included five non-professionals recruited through a crowdsourcing platform; the second category comprised three experts with extensive experience in the field of origami. Participants in these evaluations were compensated according to the prevailing local minimum hourly wage standard.

## C  Detailed Explanation of Origami Compiler Error Feedback System

The following is a detailed supplementary explanation of the origami compiler error feedback system, including more specific error types, possible error messages, relevant parameters, and their underlying principles.

### C.1  CP Code Syntax Error

This type of error occurs in the initial phase when the compiler parses the Crease Pattern (CP) code provided by the user, if the code does not conform to predefined syntax rules.

#### C.1.1  More Details

- **Example Error Codes:**
    - `E_CP_SYNTAX_INVALID_PARAM_COUNT`: "Instruction '$COMMAND$' has an insufficient or excessive number of parameters. Expected $X$, got $Y$."
    - `E_CP_SYNTAX_UNKNOWN_COMMAND`: "Unrecognized instruction '$COMMAND$'. Please check spelling or the instruction set."
    - `E_CP_SYNTAX_INVALID_PARAM_TYPE`: "Parameter '$PARAM\_NAME$' for instruction '$COMMAND$' has an invalid type. Expected type '$EXPECTED\_TYPE$', but received value '$VALUE$' of type '$ACTUAL\_TYPE$'."

- E_CP_SYNTAX_VALUE_OUT_OF_RANGE: "Value $'VALUE'$ for parameter $'PARAM\_NAME'$ of instruction $'COMMAND'$ is out of the allowed range $[MIN\_VAL, MAX\_VAL]$."
- E_CP_SYNTAX_UNEXPECTED_TOKEN: "Unexpected symbol/character $'TOKEN'$ encountered at line number $[line\_number]$, column $[col\_number]$ while parsing instruction $'COMMAND'$."
- E_CP_SYNTAX_MISSING_DELIMITER: "Instruction $'COMMAND'$ is missing a required delimiter. For example, the expected $'EXPECTED\_DELIMITER'$ was not found."
- E_CP_SYNTAX_INVALID_LINE_REFERENCE: "Instruction $'COMMAND'$ references a non-existent line ID $'LINE\_ID'$ or point ID $'POINT\_ID'$."

- **faulty_cp_code_line_numbers**: $[line\_number]$ - The specific code line where the error occurred.

- **faulty_token_or_command**: (Optional) Indicates the specific instruction or token that caused the error.

### C.1.2 Underlying Principles

- **Formal Language and Grammar:** CP code is treated as a formal language with precisely defined lexical and syntax rules.

- **Parsing Stages:**
  1. **Lexical Analysis:** Code text is broken into "tokens."
  2. **Syntax Analysis:** Token sequence is checked against grammar rules, often building an Abstract Syntax Tree (AST).

- **Error Detection:** Errors are reported if tokens or their sequence violate rules, preventing further compilation.

## C.2 Geometrically Impossible Fold

This error indicates that some defined folding operations are physically or geometrically unfeasible.

### C.2.1 More Details

- **Example Error Codes:**
  - E_GEOM_TOO_MANY_LAYERS: "Folding near $(x, y)$ would result in $N$ paper layers, exceeding the limit of $M$ layers."
    * max_allowable_layers: Maximum allowed layers.
    * calculated_layers_at_point: Calculated layers at the point.
  - E_GEOM_ANGLE_CONSTRAINT_VIOLATION: "Target angle $[\theta_{target}]$ of crease $[id]$ conflicts with existing angles $[\alpha_1, \ldots, \alpha_{2n}]$ at vertex $[vertex\_coordinates]$."
    * **Specific reasons may include:**
      · "Maekawa-Justin: $|M - V| \neq 2$."
      · "Kawasaki-Justin: $\sum(-1)^i \alpha_i \neq 0$ or alternate sums $\neq \pi$ for flat-folds."
      · "Angle sum around vertex $\neq 2\pi$ (internal) or $\pi$ (boundary)."
      · "Single crease angle is too large or small."
    * conflicting_crease_ids_and_angles: IDs and angles of conflicting creases.
  - E_GEOM_CREASE_PLACEMENT_INVALID: "Endpoints of crease $[id]$ are outside paper, or crease illegally intersects boundary."
  - E_GEOM_LENGTH_CONSTRAINT_VIOLATION: "Operation requires points $[A]$ and $[B]$ to coincide, but original distance $d_1 \neq$ required $d_2$ (usually 0), implying stretching."

- **faulty_crease_ids**: $[List of crease IDs causing the conflict]$

- **faulty_vertex_ids_or_point_coordinates**: $[Conflicting vertex IDs or point coordinates]$

- **problematic_coordinates_or_regions**: $[Problematic region's coordinates or description]$

### C.2.2 Underlying Principles

- **Non-stretchability of Paper:** Paper is inextensible; folding is an isometric transformation.
- **Local Developability:** Paper must be locally developable onto a plane (zero Gaussian curvature except at singularities).
- **Flat-foldability Conditions:** For flat folds:
  - **Maekawa-Justin Theorem:** $|M - V| = 2$.
  - **Kawasaki-Justin Theorem:** $\sum_{i=1}^{n} \alpha_{2i-1} = \sum_{i=1}^{n} \alpha_{2i} = \pi$.
  - **Big-Little-Big Angle Constraint:** $\alpha_i \leq \alpha_{i-1} + \alpha_{i+1}$.
- **Layer Thickness Limitation:** Real paper has thickness, limiting layer stacking.

## C.3 Paper Self-Intersection/Penetration

This error means different paper parts occupy the same 3D space.

### C.3.1 More Details

- **Example Error Codes:**
  - E_PHYS_SELF_INTERSECTION: "After crease $[id]$, facet $[facet\_A\_id]$ (region $[coordinates_A]$) penetrates facet $[facet\_B\_id]$ (region $[coordinates_B]$)."
  - E_PHYS_INTERSECTION_DURING_MOTION: "During folding of $[id]$, at time $t = [time]$, region $[region\_A]$ collides with $[region\_B]$."
  - E_PHYS_BOUNDARY_VIOLATION: "Folded part $[facet\_id]$ penetrates defined container boundary."
- **faulty_crease_ids**: $[CreaseID(s) causing or related to penetration]$
- **problematic_coordinates_or_regions**: $[Penetration area : pointsets, bounding boxes, or facet IDs]$
- **intersecting_layer_ids / intersecting_facet_ids**: (Optional) $[layer\_id_1, layer\_id_2]$ or $[facet\_id_1, facet\_id_2]$ specifying penetrating parts.
- **penetration_depth**: (Optional) Estimated penetration depth/volume. E.g., $d = 0.5$mm.

### C.3.2 Underlying Principles

- **Volumetric Exclusion:** Physical objects cannot occupy the same space.
- **Collision Detection:** Algorithms detect intersections between paper parts (meshes/facets).
  - **Discrete Collision Detection:** Checks static geometry at time steps.
  - **Continuous Collision Detection (CCD):** Detects collisions between time steps to prevent "tunneling."
- **Data Structures:** Spatial partitioning (Octrees, BVHs) for efficient detection.
- **Layer Ordering and Penetration:** Incorrect layer order in flat folds can cause penetration.

## C.4 Ambiguous Folding State

Indicates that the CP code and constraints do not uniquely determine the folded form.

### C.4.1 More Details

- **Example Error Codes:**
  - E_AMBIGUOUS_STATE: "CP code is insufficient for a unique state. $N$ possible configurations in region $[coordinates]$ (or vertex $[vertex\_id]$)."
  - E_AMBIGUOUS_LAYER_ORDER: "Insufficient constraints to determine stacking order of layers $[layer\_A\_id]$ and $[layer\_B\_id]$ in region $[coordinates]$. At least two valid orders."
  - E_AMBIGUOUS_TUCK_CHOICE: "Operation 'tuck' at $[coordinates]$ has multiple valid insertion methods; CP unspecified."

– E_AMBIGUOUS_MOUNTAIN_VALLEY_ASSIGNMENT: "For crease $[crease\_id]$, multiple valid M/V assignments satisfy local constraints but yield different global forms."

- **problematic_coordinates_or_regions**: $[Region or vertex where ambiguity occurs]$
- **ambiguous_crease_ids_or_vertex_ids**: (Optional) $[Crease/vertex IDs related to ambiguity]$
- **number_of_possible_states**: (Optional) Number of possible states detected ($N$).
- **suggested_disambiguation**: (Optional) "Suggestion: Add layer order constraint (e.g., LAYER_ABOVE) or specify crease direction."

### C.4.2  Underlying Principles

- **Non-uniqueness of Solution Space:** A CP may correspond to multiple valid configurations.
- **Local vs. Global Information:** Local constraints may be met, but global form can vary.
- **Symmetry:** Symmetric CPs or operations can lead to multiple equivalent results.
- **Branching Points in Configuration Space:** Folding path may have bifurcations.
- **Implicit vs. Explicit Instructions:** Unstated conventions can lead to ambiguity for the compiler.
- **Solver Behavior:** Solvers for underdetermined systems might not find a unique solution.

## D  Crease Pattern evaluation system

This section introduces the complete evaluation process of the Crease Pattern . The final score is a weighted average of the scores from the different dimensions. Each of the four main dimensions is assigned an equal weight:

- Topological Similarity: $w_{topological} = 0.25$
- Geometric Similarity: $w_{geometric} = 0.25$
- Foldability Constraint Satisfaction: $w_{foldability} = 0.25$
- Final Folded State: $w_{fold\_state} = 0.25$

The total score $S_{total}$ is calculated as:

$$S_{total} = \sum_{dim} w_{dim} \cdot s_{dim}$$

Since $\sum w_{dim} = 1$ with these weights, this simplifies to:

$$S_{total} = 0.25 \cdot s_{topological} + 0.25 \cdot s_{geometric} + 0.25 \cdot s_{foldability} + 0.25 \cdot s_{fold\_state}$$

where $s_{dim}$ is the score for a particular dimension.

### D.1  CP Structure Validation (`validate_cp_structure`)

This initial step ensures the generated CP data (`cp_data`) is well-formed and meets basic criteria for a valid crease pattern.

- **Presence of Basic Elements**: Checks if `"vertices_coords"`, `"edges_vertices"`, and `"faces_vertices"` keys exist in the input.
- **Vertex Coordinates**: Each vertex in `vertices_coords` must be a list of two numerical coordinates (e.g., `[x, y]`).
- **Edge Definitions**: Each edge in `edges_vertices` must be a list of two integer vertex indices (e.g., `[v1, v2]`). These indices must be valid and within the bounds of the vertex list.
- **Crease Assignments (Optional)**: If `"edges_assignment"` is present, each assignment must be one of the valid types: "B" (Boundary), "M" (Mountain), "V" (Valley), "F" (Flat), "U" (Unassigned).

- **Face Definitions**: Each face in `faces_vertices` must be a list of at least three integer vertex indices. These indices must be valid.
- **Euler Characteristic**: For a planar graph, the Euler characteristic must satisfy $V - E + F = 2$, where $V$ is the number of vertices, $E$ is the number of edges, and $F$ is the number of faces.
- **Flat-Folder Validation (Optional)**: If the Flat-Folder `compute` module is available, its `validate_cp_structure(cp_data)` API is called to check if the CP can be compiled into a valid origami model. If not, the CP is considered invalid.

If any of these checks fail, the function returns `{"valid": False, "reason": "error message"}`. Otherwise, it returns `{"valid": True}`.

### D.2 Topological Similarity (`calculate_topological_similarity`)

This dimension assesses the similarity of the graph-theoretical structure of the generated CP (`gen_cp`) and the reference CP (`ref_cp`). It combines scores from four sub-metrics, after extracting basic topological information using `extract_topology(cp_data)`, which retrieves vertices, edges, edge assignments, and faces.

The overall topological similarity score $S_{topological}$ is a weighted average defined within the `calculate_topological_similarity` method:

$$S_{topological} = 0.2 \cdot s_{vertex} + 0.3 \cdot s_{edge} + 0.3 \cdot s_{face} + 0.2 \cdot s_{crease}$$

### D.3 Vertex Count Similarity (`compare_vertex_count`)

Compares the number of vertices ($V_{gen}$, $V_{ref}$).

- If $V_{gen} = V_{ref}$, score $s_v = 1.0$.
- Otherwise, the score is calculated using an exponential decay function:

$$s_v = e^{-0.5 \cdot \frac{|V_{gen} - V_{ref}|}{\min(V_{gen}, V_{ref})}}$$

  (Note: The code implements this as $\exp(-0.5 \cdot (\max(V_{gen}, V_{ref}) - \min(V_{gen}, V_{ref}))/\min(V_{gen}, V_{ref}))$.)

### D.4 Edge Connectivity Similarity (`compare_edge_connectivity`)

Compares the edge structures based on degree distribution and connected components.

- **Adjacency List Construction** (`build_adjacency_list`): Adjacency lists are built for both CPs from their edge-vertex relationships.
- **Degree Distribution Similarity**:
  - `calculate_degree_distribution`: Computes the distribution of vertex degrees (number of edges connected to each vertex).
  - `calculate_wasserstein_distance`: A simplified Wasserstein distance ($d_W$) is calculated between the degree distributions of the generated and reference CPs. The score for degree similarity is $s_{degree} = 1 - d_W$.
- **Connected Components Similarity**:
  - `count_connected_components`: The number of connected components ($C_{gen}$, $C_{ref}$) is determined for each CP graph using Depth First Search (DFS).
  - If $C_{gen} = C_{ref}$, $s_{conn} = 1.0$.
  - Otherwise, $s_{conn} = e^{-|C_{gen} - C_{ref}|}$.
- The final edge connectivity score $s_{edge}$ is a weighted average: $s_{edge} = 0.7 \cdot s_{degree} + 0.3 \cdot s_{conn}$.

**D.5 Face Relations Similarity (`compare_face_relations`)**

Compares properties of the faces in the two CPs.

- **Face Count Similarity** ($s_{f\_count}$):

$$s_{f\_count} = e^{-\frac{|F_{gen}-F_{ref}|}{\max(1,\min(F_{gen},F_{ref}))}}$$

   where $F_{gen}$ and $F_{ref}$ are the number of faces.
- **Average Vertices per Face Similarity** ($s_{f\_avg\_v}$): Let $avgV_{gen}$ and $avgV_{ref}$ be the average number of vertices per face.

$$s_{f\_avg\_v} = e^{-\frac{|avgV_{gen}-avgV_{ref}|}{\max(1,\min(avgV_{gen},avgV_{ref}))}}$$

- **Face Size Distribution Similarity** ($s_{f\_dist}$): The distribution of face sizes (number of vertices per face) is computed for both CPs. A simplified Wasserstein distance ($d_W$) is calculated between these distributions using `calculate_wasserstein_distance`. The score is $s_{f\_dist} = 1 - d_W$.
- The final face relations score $s_{face}$ is a weighted average: $s_{face} = 0.3 \cdot s_{f\_count} + 0.3 \cdot s_{f\_avg\_v} + 0.4 \cdot s_{f\_dist}$.

**D.6 Crease Assignment Similarity (`compare_crease_assignment`)**

Compares the distribution of crease types (M, V, B) if `"edges_assignment"` is available.

- If either CP lacks edge assignments, a low score of $0.2$ is returned.
- **Crease Type Counts** (`count_crease_types`): Counts the occurrences of Mountain ('M'), Valley ('V'), Boundary ('B'), Flat ('F'), and Unassigned ('U') creases.
- **Proportion Similarity**: For Mountain, Valley, and Boundary creases, the similarity of their proportions ($prop$) in the generated ($gen$) and reference ($ref$) CPs is calculated:
    - Mountain: $s_M = 1 - |\text{prop}_{M,gen} - \text{prop}_{M,ref}|$
    - Valley: $s_V = 1 - |\text{prop}_{V,gen} - \text{prop}_{V,ref}|$
    - Boundary: $s_B = 1 - |\text{prop}_{B,gen} - \text{prop}_{B,ref}|$

   where proportion is count of type / total number of assigned edges for that CP.
- **Length Penalty** ($p_L$): A penalty is applied if the total number of assigned edges differs:

$$p_L = \frac{\min(L_{gen}, L_{ref})}{\max(L_{gen}, L_{ref})}$$

   where $L$ is the total number of assigned edges.
- The final crease assignment score $s_{crease}$ is a weighted average of the proportion scores, multiplied by the length penalty:

$$s_{crease} = (0.4 \cdot s_M + 0.4 \cdot s_V + 0.2 \cdot s_B) \cdot p_L$$

**D.7 Geometric Similarity (`calculate_geometric_similarity`)**

This dimension evaluates the similarity of the spatial characteristics of the compiled/folded models. It requires compiling the CPs into 3D models.

- **CP Compilation** (`compile_cp_to_model`):
    - If the Flat-Folder `compute.compute_folded_state(cp_data)` API is available, it's used to get the folded model data (typically including 3D vertex coordinates `"P"` and crease edges `"SP"`).
    - If Flat-Folder is unavailable, a `simplified_folding` method is used, which essentially returns the original 2D vertex coordinates as `"P"` and edges as `"SP"`. This is a significant simplification.

- If either CP fails to compile (or provide simplified data), a low score of $0.2$ is returned by `calculate_geometric_similarity`.

The overall geometric similarity score $S_{geometric}$ is a weighted average defined within `calculate_geometric_similarity`:

$$S_{geometric} = 0.4 \cdot s_{point} + 0.3 \cdot s_{angle} + 0.3 \cdot s_{size}$$

### D.8 Point Position Similarity (`compare_point_positions`)

Compares the 3D point clouds of the folded models.

- **Coordinate Normalization** (`normalize_coordinates`): Vertex coordinates (from `"P"`) of both models are normalized. If points are 2D, a Z-coordinate of 0 is added. Points are then translated so their centroid is at the origin and scaled so the maximum distance from the origin to any point is 1 (i.e., normalized to a unit sphere).
- **Bidirectional Hausdorff Distance** (`calculate_bidirectional_hausdorff`): The Hausdorff distance $d_H(A, B) = \max\left(\sup_{a \in A} \inf_{b \in B} d(a, b), \sup_{b \in B} \inf_{a \in A} d(a, b)\right)$ is calculated between the normalized point sets of the generated ($P_{gen}$) and reference ($P_{ref}$) models. $d(a, b)$ is the Euclidean distance. This is achieved by calling `calculate_hausdorff_distance` twice.
- The point position similarity score $s_{point}$ is calculated using an exponential decay function:

$$s_{point} = e^{-k \cdot d_H}$$

  where $k = 5$ is a sensitivity coefficient.

### D.9 Angle Similarity (`compare_angles`)

Compares the distribution of dihedral angles along creases in the folded models.

- **Crease Edge Extraction** (`extract_crease_edges`): Crease edges are extracted from the folded model data (typically from `"SP"`).
- **Dihedral Angle Calculation** (`calculate_dihedral_angles`):
  - **Note**: In the provided `eval.py`, if Flat-Folder is unavailable, this function returns a list of *random angles* as a placeholder. A proper implementation would calculate actual dihedral angles between faces sharing a crease.
- **Angle Histogram Comparison** (`compare_angle_histograms`):
  - `create_histogram`: Histograms of dihedral angles are created for both models. Angles are typically in $[0, 180°]$, binned into 18 bins (10 degrees per bin).
  - `calculate_cosine_similarity`: The cosine similarity between the two angle histogram vectors is calculated. This value serves as the angle similarity score $s_{angle}$.
- If creases cannot be extracted or angles cannot be calculated for either model, a default score of $0.5$ is returned by `compare_angles`.

### D.10 Size and Proportions Similarity (`compare_size_and_proportions`)

Compares the overall dimensions and aspect ratios of the folded models' bounding boxes.

- **Bounding Box Calculation** (`calculate_bounding_box`): The axis-aligned bounding box (min/max coordinates along X, Y, Z) is computed for the point clouds of both models. 2D points are padded with Z=0.
- **Proportion Calculation**: The dimensions (length, width, height) of the bounding boxes are calculated. These dimensions are sorted in descending order and then normalized by dividing by the largest dimension (e.g., $[1, L_2/L_1, L_3/L_1]$).
- **Similarity Score**: The cosine similarity between the normalized proportion vectors of the two models is calculated using `calculate_cosine_similarity`. This value is the size and proportions similarity score $s_{size}$.
- If either point set is empty, a default score of $0.5$ is returned by `compare_size_and_proportions`.

### D.11 Foldability Constraint Satisfaction (`calculate_foldability_similarity`)

This dimension assesses whether the generated CP adheres to known origami foldability constraints, beyond basic geometric foldability.

- **Basic Foldability Check (Optional)**:
  - If Flat-Folder's `compute.check_foldability(cp_data)` API is available, it's used to check if both CPs are foldable.
  - If the reference CP is foldable but the generated CP is not, the score for `calculate_foldability_similarity` returns $0.2$.

The overall foldability score $S_{foldability}$ is a weighted average defined within `calculate_foldability_similarity`:

$$S_{foldability} = 0.3 \cdot s_{TT} + 0.3 \cdot s_{TTo} + 0.2 \cdot s_{Trans} + 0.2 \cdot s_{flatfold}$$

If an exception occurs during calculation, `calculate_foldability_similarity` returns a score of $0.3$.

### D.12 Specific Origami Constraint Comparison

This involves extracting and comparing critical origami constraints.

- **Constraint Extraction** (`extract_constraints`):
  - This method aims to extract Taco-Taco (TT), Taco-Tortilla (TTo), and Transitivity (Trans) constraints by calling helper methods like `extract_taco_taco_constraints`.
  - **Note**: In the provided `eval.py`, if Flat-Folder's `constraints` module is unavailable, the extraction methods are simplified and return empty lists. A full implementation would identify these constraints from the CP geometry and crease assignments.

- **Constraint Set Comparison** (`compare_taco_taco_constraints`, `compare_taco_tortilla_constraints`, `compare_transitivity_constraints`): For each constraint type (TT, TTo, Trans):
  - If both CPs have no such constraints, similarity is $1.0$.
  - If one has constraints and the other doesn't, similarity is $0.3$.
  - Otherwise:
    * **Constraint Overlap** ($s_{overlap}$): Calculated using Jaccard similarity on the sets of constraints (constraints are stringified for comparison via `calculate_constraint_overlap`).

    $$J(A, B) = \frac{|A \cap B|}{|A \cup B|}$$

    * **Count Similarity** ($s_{count}$):

    $$s_{count} = e^{-\frac{|N_{gen} - N_{ref}|}{\max(1, \min(N_{gen}, N_{ref}))}}$$

    where $N$ is the number of constraints of that type.
    * The score for that constraint type (e.g., $s_{TT}$) is $0.7 \cdot s_{overlap} + 0.3 \cdot s_{count}$.

### D.13 Local Flat-Foldability Conditions (`compare_flat_foldability`)

Checks for adherence to local flat-folding theorems around vertices.

- **Kawasaki's Theorem Check** (`check_kawasaki_theorem`):
  - States that for a flat-foldable vertex, the sum of alternating angles around the vertex is $180°$, or equivalently, $\sum \alpha_i = 2\pi$ (or 0, depending on how angles are measured like $\sum (-1)^i \alpha_i = 0$).

- **Note**: The mock implementation in `eval.py` always returns `True`. A full implementation would iterate internal vertices and check angles.
- **Maekawa's Theorem Check** (`check_maekawa_theorem`):
  - States that for a flat-foldable vertex, the number of mountain creases ($M$) and valley creases ($V$) must differ by two: $|M - V| = 2$.
  - **Note**: The mock implementation in `eval.py` always returns `True`. A full implementation would check crease assignments around internal vertices.
- **Scoring**:
  - Kawasaki score ($s_K$): $0.2$ if reference theorem status is True and generated is False, $1.0$ otherwise.
  - Maekawa score ($s_M$): $0.2$ if reference theorem status is True and generated is False, $1.0$ otherwise.
- The final flat-foldability score $s_{flatfold} = 0.5 \cdot s_K + 0.5 \cdot s_M$.

### D.14 Final Folded State Similarity (`compare_final_folded_state`)

This dimension directly compares the 3D geometry of the final folded shapes compiled from the generated and reference CPs.

- **CP Compilation**: Similar to geometric similarity, `compile_cp_to_model` is used. If compilation fails for either (returns falsy), `compare_final_folded_state` returns a score of $0.3$.
- **Point Cloud Extraction**: 3D vertex coordinates ("P") are extracted from the compiled models. If point clouds are missing for either, a score of $0.3$ is returned.

The overall final folded state score $S_{final\_state}$ is a weighted average defined within `compare_final_folded_state`:

$$S_{final\_state} = 0.7 \cdot s_{shape} + 0.3 \cdot s_{layer}$$

If an exception occurs during calculation, `compare_final_folded_state` returns $0.3$.

### D.15 Overall Shape Similarity

- Calculated using the bidirectional Hausdorff distance $d_H$ between the (normalized) point clouds of the generated and reference folded models, identical to the method in `compare_point_positions`.
- The shape similarity score $s_{shape}$ is:

$$s_{shape} = e^{-5 \cdot d_H}$$

### D.16 Layering Similarity (`compare_layers`)

Compares the stacking order of faces/layers in the folded state.

- This relies on layering information being present in the compiled model, typically under a key like "CF" (face assignments or configuration).
- **Note**: The `compare_layers` function in the provided `eval.py` is a simplified placeholder and returns a default score of $0.5$. A full implementation would require a detailed comparison of the layer graph or face ordering.
- The score is $s_{layer}$.

## E  Training setting

For the reinforcement learning method, we adopt TRICO [35] for training on qwen2.5-vl-32B, which is a PPO-based [36], more efficient MLLMs multi-turn reinforcement learning algorithm. Specifically, we trained for 10.2 hours on 16 H100 GPUs, with the following hyperparameter settings: $\gamma_{turn} = 0.95$, $\gamma_{token} = 1.0$, KL penalty $= 0.001$, Actor LR$=1 \times 10^{-6}$, and Critic LR$=1 \times 10^{-5}$.

# F Limitation

While the ORIGAMISPACE benchmark and dataset offer a novel approach to evaluating multi-step spatial reasoning in MLLMs, we acknowledge certain limitations that provide avenues for future work. Firstly, although our dataset comprises 350 meticulously collected origami instances, the overall scale is relatively modest compared to some large-scale benchmarks in other vision and language domains. Future efforts could focus on expanding the dataset size and further diversifying the range of origami types and complexities included, potentially through semi-automated generation techniques, to ensure even broader coverage and statistical power. Secondly, while origami provides an excellent structured environment with clear mathematical constraints, the direct transferability of MLLM performance and the specific reasoning mechanisms learned on ORIGAMISPACE to other, less constrained or visually distinct spatial reasoning tasks (e.g., understanding dynamic real-world scenes or interpreting abstract diagrams from different fields) warrants further investigation. Exploring this generalization gap could be a valuable direction for future research. Finally, our current set of evaluation tasks, though designed to be challenging, focuses on specific facets of spatial reasoning highlighted by origami. There may be other subtle aspects of spatial intelligence or different interaction modalities with the origami compilation process that could be explored in future iterations to provide an even more holistic assessment of MLLM capabilities.

