# OpenReview forum: "ORIGAMISPACE: Benchmarking Multimodal LLMs in Multi-Step Spatial Reasoning with Mathematical Constraints"
_NeurIPS.cc/2025/Conference — NeurIPS 2025 spotlight_

### Official Review · Reviewer_GYhc · 2025-07-01

**Clarity:** 3
**Significance:** 3
**Originality:** 4
**Rating:** 5
**Confidence:** 3

**Summary:**

This paper introduces ORIGAMISPACE, a novel benchmark designed to evaluate the multi-step spatial reasoning capabilities of Multimodal Large Language Models (MLLMs), with a particular focus on their ability to adhere to precise mathematical constraints. The authors argue that existing benchmarks often focus on static scenes or simple spatial transformations, lacking the complexity of sequential, constrained, and dynamic reasoning found in real-world tasks. To address this gap, they propose using the domain of origami, which inherently involves a sequence of dependent folding operations governed by strict geometric rules.

Experiments on a wide range of state-of-the-art open and closed-source MLLMs (including GPT-4o and Gemini 2.5-pro) demonstrate that ORIGAMISPACE is highly challenging. All models perform significantly below human expert levels, and the results highlight the profound difficulty MLLMs have with fine-grained geometric reasoning and constraint satisfaction. The work also shows that interactive learning with compiler feedback dramatically improves performance over standard in-context learning.

**Questions:**

Dataset Scale and Robustness: The dataset size of 350 test instances is modest for a benchmark of this nature. Could the authors comment on the statistical significance of the reported performance gaps between models? Have you considered or are you planning methods to scale the dataset, perhaps through procedural generation of simpler origami CPs and their corresponding folded forms, to further improve the benchmark's robustness?

Input to Code Generation Task: For the End-to-End CP Code Generation task (Task 4), the input is described as a "compiled flat layout and an image of the folded shape." Providing the compiled flat layout seems to give the model a significant amount of information that is not typically available in a real-world "reverse engineering" scenario. Could you clarify the motivation for this choice? Would it not be a more challenging and perhaps more realistic task to generate the CP code from only the folded shape image, possibly augmented with natural language instructions?

Generalization of RL Agent: The reinforcement learning results are very promising, showing that an agent can learn to interact with the compiler to improve its output. Could you provide more insight into the generalization capabilities of the trained Qwen-VL-32B agent? For instance, how well does it perform on origami designs of a complexity level or type not seen during its training? Is the learned policy mostly correcting minor syntax errors, or is it demonstrating a deeper, more generalizable understanding of geometric constraints?
Compiler and Environment Availability: A benchmark's utility is greatly enhanced by its accessibility. Will the optimized origami compiler, the interactive environment, and the full evaluation suite be made publicly available alongside the dataset?

A positive response to these questions, particularly clarifying the design choice in Question 2 and discussing the potential for scaling in Question 1, would further strengthen this already excellent paper.

**Ethical Concerns:**

["NO or VERY MINOR ethics concerns only"]

**Limitations:**

Yes, the authors have adequately addressed the limitations of their work in Appendix F. They correctly identify the modest dataset scale, the open question of transferability to other domains, and the potential to expand the suite of evaluation tasks as avenues for future work. This is a thoughtful and honest assessment.

**Paper Formatting Concerns:**

None. The paper adheres to the NeurIPS formatting guidelines.

**Quality:**

3

**Strengths And Weaknesses:**

Strengths:

Significance and Originality: The paper identifies a clear and critical weakness in current MLLM evaluation: the lack of benchmarks for multi-step, mathematically-grounded spatial reasoning. The choice of origami as a domain is both highly original and exceptionally well-suited to this problem. It provides a task that is intuitive to understand yet programmatically verifiable, procedurally complex, and rich with implicit and explicit constraints. This is a far more rigorous test of spatial "imagination" than what is offered by most existing benchmarks.

Quality of the Benchmark Design: The benchmark is exceptionally well-designed.

The four tasks are diverse, probing different facets of spatial reasoning from high-level pattern recognition to low-level geometric prediction and generative construction.

The use of an origami compiler is a major strength. It provides a formal, non-negotiable ground truth for evaluation, which is a significant step up from relying on human consensus or simple metric comparisons. The detailed error feedback system (CSE, GIF, PSI, AFS) is very well-thought-out and enables the novel interactive evaluation setting.

The evaluation metrics for the CP code generation task (TSS, GS, CS, FFS) are comprehensive, covering topological structure, geometric similarity, constraint satisfaction, and the final folded state. The level of detail in Appendix D is commendable and demonstrates a deep engagement with the problem.

Quality of the Experimental Evaluation: The empirical study is thorough and compelling.

The authors evaluate a wide and very current set of both open-source and proprietary SOTA MLLMs, making the results highly relevant.
The inclusion of both layperson and expert human performance provides essential context and clearly demonstrates the difficulty of the tasks and the significant gap that remains for AI.

The comparison between in-context learning, environmental learning (with interaction), and reinforcement learning provides a key insight: that iterative refinement based on structured feedback is a powerful and necessary paradigm for solving such complex tasks.

Clarity and Reproducibility: The paper is extremely well-written, logically structured, and easy to follow. The extensive appendices are a model of good practice; they provide exhaustive detail on the dataset, annotation rules, compiler errors, and evaluation metrics, which will be invaluable for any researcher wishing to build upon this work.

Weaknesses:

Dataset Scale: The primary weakness is the relatively modest scale of the dataset (350 test instances, with an additional 471 used for RL training). While the quality of each instance is clearly very high due to meticulous curation, this size might raise concerns about statistical power and the potential for models to overfit, particularly in the reinforcement learning setting.

Scope and Transferability: While origami is an excellent proxy for constrained spatial reasoning, the paper could benefit from a slightly broader discussion on the expected transferability of skills developed on ORIGAMISPACE to other domains like robotics (e.g., cloth folding, assembly), molecular biology (protein folding), or architectural design. This is briefly acknowledged in the limitations but could be expanded upon in the main text to strengthen the paper's impact statement.

---

> ### Author Rebuttal · Authors · 2025-07-30
>
> We sincerely appreciate your thorough review and valuable suggestions. We have provided detailed responses to all of your comments and questions. We hope that our clarifications will help you to better understand the specifics of our work and findings.
>
> > Regarding Dataset Scale and Robustness
>
> Regarding the statistical significance of the performance differences between models, we report in Table 1 the average of three independent runs, and each model's performance score is appended with a "±" value. Regarding the data regularity, thank you for your valuable suggestions. This is indeed a focus of our future work.
>
> We have attempted semi-automated methods but found that the usability of CP (Crease Pattern) generated by the model is extremely low when reviewed by humans (success rate less than 1%). The cost of using the model to generate and then having humans verify is also very high. We believe that the current MLLM's capabilities are not yet sufficient to handle high-quality semi-automated generation tasks, which will require continuous iteration and optimization of the model version to be realized in the future.
>
> Therefore, our current strategy is to collaborate with established origami communities and databases. We have reached cooperation intentions with creators from several resource websites, including Global Origami Community and Origami Database. We plan to officially cooperate with them to bring in a larger volume of origami model data that has been verified and organized by humans. This approach not only ensures the quality and accuracy of the data but also allows us to expand the dataset more efficiently on the basis of the existing 350 examples.
>
> Looking forward, we believe these two methods are complementary. With the improvement of MLLM capabilities, we will re-examine and deploy semi-automated generation processes, which will allow our community experts to participate in the "human-in-the-loop" process, screening and optimizing the procedurally generated models, thereby balancing efficiency and quality.
>
> > Input to Code Generation Task
>
> We agree with your point of view. Providing the model with a "Compiled Flat Pattern" compared to a "Folded Shape Image" does indeed reduce the "real-world" inverse engineering difficulty of the task. However, our goal is to provide a clearer final result, because a "Folded Shape Image" is usually a photo or a 3D render, which itself carries a large amount of **perceptual uncertainty**, such as viewpoint, lighting, shadows, material texture, and non-rigid deformation. Requiring a model to directly generate precise CP code from such an image would mix the core **spatial reasoning** challenges with complex **3D visual perception and reconstruction** challenges.
>
> We completely agree with your view that generating CP code from a "Folded Shape Image" (possibly with natural language instructions) is a more challenging and more intuitive inverse engineering task. We believe this represents a more difficult and more valuable future research direction.
>
> Our ORIGAMISPACE benchmark can be seen as a **key intermediate step** in achieving this grand goal. It pioneers the systematic evaluation and improvement of the ability of Multi-modal Large Language Models (MLLM) in core **geometric constraints and multi-step spatial reasoning** in a well-controlled, perception-free environment. Once a model demonstrates sufficient capability on this foundational task, the next step is to challenge more open and realistic ultimate tasks.
>
> > Generalization of RL Agent
>
> Thank you for pointing this out. Regarding generalization, our experimental setup itself is a kind of test. The RL agent is trained on a dataset containing 471 examples. The final results (Table 2) are evaluated on the ORIGAMISPACE benchmark test set, which contains 350 examples. All the origami designs in the test set are **completely new** to the RL agent during the training process.
>
> Regarding the strategy, in the early stages of training, the model's compilation success rate (CPR) is low. The main feedback comes from the negative penalty for compilation failures. During the middle stage of training, the compilation-related rewards increase rapidly, especially for CP Syntax Error (CSE) and Geometrically Impossible Fold (GIF). We believe this is because these two types of constraints are relatively simple, allowing the model to learn to pass compilation relatively well. In the later stages, the model's CPR gradually stabilizes, but the code quality score, especially the topological similarity (TSS), shows a more noticeable increase, while the geometric similarity (GS) does not change much. This may be because the feedback for the latter is not well-designed, and the required capabilities are more abstract. Furthermore, constrained by the penalty for step length, the number of interaction rounds required to complete the task will decrease. Overall, the model will first learn simpler strategies, such as Euler's formula for planar graphs and Maekawa's theorem, and then try to learn more complex strategies.
>
> > Accessibility of Compilers and Environments
>
> Yes, we will publicly release all data, environments (compilers), training and evaluation code, and models in the short term.
>
> Thank you again for your recognition of our work, as well as for your constructive comments and suggestions!

---

> ### Comment · Area_Chair_aVQX · 2025-08-03
>
> Dear Reviewer,
>
> Could you please check if the authors’ rebuttal adequately addresses your concerns? If so, kindly acknowledge the rebuttal and provide any additional comments. If not, it would be greatly appreciated it if you could engage in a discussion with the authors. Your input at this stage is essential to the review process. Thank you very much for your time and effort!
>
> AC

---

> ### Comment · Area_Chair_aVQX · 2025-08-06
>
> Dear Reviewer,
>
> According to this year's NeurIPS review policy, "*Reviewers must participate in discussions with authors before submitting a Mandatory Acknowledgement*", could you please provide additional comments discussing whether the rebuttal addresses your concerns?
>
> Thank you.
>
> AC

---

### Official Review · Reviewer_mktL · 2025-07-02

**Clarity:** 3
**Significance:** 3
**Originality:** 3
**Rating:** 5
**Confidence:** 2

**Summary:**

This paper introduces a spatial reasoning benchmark for MLLMs grounded in origami folding. The benchmark defines four tasks: final shape prediction, multistep spatial reasoning, spatial relationship prediction, and CP Code prediction, which symbolically encodes crease patterns for origami compilation. The authors also introduce four evaluation directions for CP Code quality and use the benchmark to evaluate several foundational models using in-context learning, environment learning, and reinforcement learning.

**Questions:**

Questions can be found in the weakness section.

**Ethical Concerns:**

["NO or VERY MINOR ethics concerns only"]

**Final Justification:**

I raised some questions regarding missing relevant comparisons and more analysis of the parts of the presented benchmark. The authors' response addresses the questions satisfactorily. And I have raised my rating to "Accept".

**Quality:**

3

**Strengths And Weaknesses:**

Strengths

The use of origami as a grounding mechanism for spatial reasoning is novel and well-motivated. It provides a mathematically constrained yet visually interpretable setup for multistep reasoning, which avoids the overly synthetic feel of many spatial datasets. The benchmark is thoughtfully constructed with attention to dataset realism and bias reduction. The paper also explores multiple prediction setups (in-context, environment learning, RL), which makes the evaluation broad and informative for understanding MLLM capabilities across paradigms.

Weaknesses

1. The authors are urged to add OpenAI’s recent reasoning-capable multimodal models (o3)  here, given their strong performance on comparable tasks. Even a qualitative comparison would be useful.

2.  Explanations for what each crease type (mountain, valley, etc.) does or what the full syntax rules are for compiling a valid code are unclear This makes the CP Code prediction task hard to interpret. The paper would benefit from a more detailed breakdown or a concrete example of CP Code semantics.

---

> ### Author Rebuttal · Authors · 2025-07-30
>
> We sincerely appreciate your thorough review and valuable suggestions. We have provided detailed responses to all of your comments and questions. We hope that our clarifications will help you to better understand the specifics of our work and findings.
>
> > Given that OpenAI's recent multimodal model (o3), which has reasoning capabilities, has performed exceptionally well in similar tasks, it is recommended that the authors include this model here. Even a qualitative comparison would be helpful.
>
> Thank you for pointing this out. We have added the experimental results of o3 on different spatial reasoning tasks, as shown in the tables below:
>
> | Pattern Prediction | Multi-step Spatial Reasoning | Spatial Pose Localization | Layering Relationship | Geometric Change |
> |:------------------:|:----------------------------:|:-------------------------:|:---------------------:|:----------------:|
> | 44.01 | 52.89 | 49.23 | 51.86 | 46.95 |
>
> | | **Compilation** | | | | | **Quality** | | | | |
> |:---|:---:|:---:|:---:|:---:|:---:|:---:|:---:|:---:|:---:|:---:|
> | **Setting** | **CSE** | **GIF** | **PSI** | **AFS** | **CPR** | **TSS** | **GS** | **CS** | **FFS** | **Total** |
> | **In-context learning** | 94.27 | **62.55** | 51.26 | **46.95** | **31.12** | **52.09** | 43.26 | **44.87** | **39.11** | **44.20** |
> | **Environmental learning** | **100** | **93.51** | 91.29 | 83.50 | **69.44** | **61.75** | 51.88 | 55.08 | **46.92** | 53.65 |
>
> As can be seen, o3 performs very well on ORIGAMISPACE, comparable to gemini-2.5-pro. We will add these results in the next version.
>
> > The function of each crease type (mountain fold, valley fold, etc.) and the complete syntax rules for writing valid code are not clearly explained. This makes the CP code prediction task difficult to understand. The paper would be more helpful if it could break down the semantics of CP code in more detail or provide a specific example.
>
> Certainly. Below is an introduction to the syntax rules for CP code:
>
> A complete CP code consists of four parts: vertex coordinates (`vertices_coords`), edge-to-vertex relationships (`edges_vertices`), edge type assignments (`edges_assignment`), and face-to-vertex relationships (`faces_vertices`).
>
> * **`vertices_coords`**: This defines the positions of all key points on the paper, represented by the 2D coordinates of each point.
> * **`edges_vertices`**: This defines the line segments, including the paper's boundaries and the internal creases. Each edge is defined by connecting the indices of two vertices.
> * **`edges_assignment`**: This is the most critical step, representing the property of each "edge." This includes "B" (Boundary), "M" (Mountain fold), "V" (Valley fold), and "F" (Flat fold).
> * **`faces_vertices`**: This defines all the 2D "faces" that are partitioned by the edges and creases. Each face is defined by the sequence of vertices that make up its contour.
>
> In the next version of the paper, we will provide a more detailed explanation of the CP code, along with an analysis of specific examples.
>
> Thank you again for your recognition of our work, as well as for your constructive comments and suggestions!

---

> > ### Comment · Reviewer_mktL · 2025-08-04
> > **Thanks for the updates. Increasing my rating.**
> >
> > I thank the authors for clarifications. I will increase my rating to "Accept".

---

> ### Comment · Area_Chair_aVQX · 2025-08-03
>
> Dear Reviewer,
>
> Could you please check if the authors’ rebuttal adequately addresses your concerns? If so, kindly acknowledge the rebuttal and provide any additional comments. If not, it would be greatly appreciated it if you could engage in a discussion with the authors. Your input at this stage is essential to the review process. Thank you very much for your time and effort!
>
> AC

---

### Official Review · Reviewer_VSem · 2025-07-03

**Clarity:** 3
**Significance:** 3
**Originality:** 3
**Rating:** 5
**Confidence:** 3

**Summary:**

This paper introduces ORIGAMISPACE, a novel benchmark dataset aimed at evaluating the multi-step spatial reasoning capabilities of multimodal large language models (MLLMs), particularly under mathematical and geometric constraints. The benchmark leverages origami-based tasks and consists of 350 instances, each including rich multimodal artifacts such as crease patterns, flat patterns, folding sequences, and final folded shapes. The authors define four tasks: Pattern Prediction, Multi-step Spatial Reasoning, Spatial Relationship Prediction and End-to-End CP Code Generation. They also propose an interactive environment for CP code generation and explore reinforcement learning (RL) techniques to enhance model training.

**Questions:**

1. Could you explain further how the mathematical constrains contribution in the proposed multi-step geometric and layering reasoning?
2. What is the impact of mathematical constrains? During each step, will be possible of precisely satisfying all mathematical constraints? if not, what is the meaning of adding these constrains?
3. Could you elaborate how RL is set up in the experiments?

**Ethical Concerns:**

["NO or VERY MINOR ethics concerns only"]

**Final Justification:**

Thank you for efforts made during Rebuttal. I am quite satisfied with the revision and explanation given, especially the importance of the math constraints and RL details.

**Limitations:**

1. data sets limitations.
2. Error analysis

**Paper Formatting Concerns:**

NA.

**Quality:**

3

**Strengths And Weaknesses:**

Strengths:
1. The use of origami as a testbed is creative and well-motivated. It naturally involves complex spatial transformations, offering a rich ground for evaluating MLLMs beyond typical vision-language tasks.
2. The four distinct tasks reflect a thoughtful design that targets various dimensions of spatial reasoning.
3. Mathmatical constraint is a key strength compared to existing loosely defined visual reasoning.

Weaknesses:
1. The dataset contains only 350 instances, which may limit statistical robustness and hinder the training of larger models.
2. The mention of RL is promising, but it fails to offer details on the task structure, reward design, or preliminary performance. It is hard to evaluate the feasibility and contributions of RL here.
3. The mathematical constraints are not well illustrated in the paper. I think it is one of you biggest contributions while the papers fails to convince reader how these constraints affect the overall performance of this problem.

---

> ### Author Rebuttal · Authors · 2025-07-29
>
> We sincerely appreciate your thorough review and valuable suggestions. We have provided detailed responses to all of your comments and questions. We hope that our clarifications will help you to better understand the specifics of our work and findings.
>
> > The dataset contains only 350 samples, which may limit statistical robustness and hinder the training of larger models.
>
> We believe that for this task, the quality of the data is more important than the quantity. The process of origami is extremely rigorous, especially since it requires strict compilation via CP code, where even a minor error can lead to a vastly different result. We previously attempted a semi-automated approach with manual filtering, but the success rate was very low.
>
> Therefore, we have partnered with established origami communities and databases. We plan to access their larger-scale, manually verified, and curated origami model data through official collaborations. This approach not only ensures data quality and accuracy but also allows us to expand our dataset more efficiently beyond the current 350 instances.
>
> > Regarding Mathematical Constraints
>
> Certainly, we will further elaborate on the impact and function of mathematical constraints to hopefully resolve your questions.
>
> First, you can think of mathematical constraints in two forms. The first is **"hard constraints,"** as described in Section 3.2 of the paper. These include CP Code Syntax Error (CSE), Geometrically Impossible Fold (GIF), Paper Self-Intersection (PSI), and Ambiguous Fold State (AFS). These four hard constraints must be satisfied; otherwise, the compilation will fail. The second is **"soft constraints,"** which refers to the Constraint Satisfaction mentioned in Section 4.4. At this stage, when the CP Code is compilable, the constraints evaluate the ability to achieve more detailed and self-consistent spatial intelligence. This primarily includes two parts: the verification of specific advanced origami constraints and local flat-foldability conditions, as described in Appendices D.11, D.12, and D.13. The former includes Taco-Taco (TT), Taco-Tortilla (TTo), and Transitivity constraints. During evaluation, the system extracts these advanced constraints from both the generated model and the reference model and then compares the similarity of the two sets. The similarity is calculated as a weighted sum of the overlap of the constraint sets (Jaccard similarity) and the similarity of the constraint counts. The latter assesses a more fine-grained folding state through local vertex metrics.
>
> To make this easier to understand, here are two examples:
>
> * **Paper Self-Intersection (PSI) - Hard Constraint:** This violates the physical principle that "different parts of an object cannot occupy the same space simultaneously." When the compiler deduces the relative positions and stacking order of the paper's parts, it will report an error if it finds any regions that would overlap or penetrate.
> * **Taco-Tortilla (TTo) - Soft Constraint:** "Taco" refers to a folded flap of paper, while "Tortilla" represents a flat, unfolded surface. The Taco-Tortilla constraint is used to specify this "one-folded, one-flat" stacking order. This decision is critical to the final model's appearance and function. For instance, it might determine whether a pocket is on the outside or inside of a garment, or whether a part of a model is convex or concave. A model that correctly handles TTo constraints demonstrates that it understands more refined spatial operations like "hiding" and "covering."
>
> Now, to answer your two questions directly:
>
> **How exactly do the mathematical constraints work?**
>
> You can understand it as the compiler performing complex mathematical derivations to calculate whether the CP code can conform to various constraint states and then returning relevant information. This provides a clear, computable feedback signal and evaluation result.
>
> **What is the impact of the mathematical constraints? Is it possible to satisfy all of them at each step?**
>
> * The impact of **hard constraints** is that a CP diagram that fails to meet them cannot be folded at all.
> * The impact of **soft constraints** is that a CP diagram that fails to meet them will have an incorrect folding state in some spatial areas compared to the reference.
>
> The role of hard constraints is to assess whether the current state is foldable. The role of soft constraints is to provide a partial reward signal *after* foldability is confirmed. Therefore, in a multi-step folding process, it is not necessary to satisfy all constraints at every step, as adjustments can be made based on the compiler's feedback. The significance of the constraints is twofold: 1. They provide our evaluation with clear standards and environmental feedback. 2. They provide a simulation that is perfectly consistent with real-world physics.
>
> > Reinforcement Learning (RL) Setup
>
> Of course. We will provide more details and plan to open-source all our data, environment, training code, and models in the short term.
>
> **Task Structure**
>
> * The agent (MLLM) engages in iterative, multi-turn interactions with the environment (the compiler). The compiler returns either a text-based success message or specific error feedback.
> * The model first performs planning to generate initial CP code, then reasons through interaction with the environment and decides on the next action, such as adding or deleting a crease.
> * This "generate-feedback-modify" loop continues for a maximum of 10 interaction rounds, as set in our experiments.
>
> **Training Algorithm & Data**
>
> * We employed the **TRICO algorithm**, an efficient multi-turn reinforcement learning algorithm based on PPO, designed for MLLMs.
> * The training data consisted of 471 independent origami datasets.
>
> **Training Hyperparameters**
>
> * Appendix E provides detailed training parameters, including: training for **10.2 hours on 16 H100 GPUs**, with Actor and Critic learning rates of **1×10⁻⁶** and **1×10⁻⁵**, respectively, among other relevant parameters.
>
> **Performance Change**
>
> * In the **early stages of training**, the model's compilation success rate (CPR) is low, with the primary feedback being a fixed negative penalty for compilation failures.
> * In the **middle phase**, compilation-related rewards increase rapidly, especially for avoiding CP Code Syntax Error (CSE) and Geometrically Impossible Fold (GIF). We believe this is because these two constraints are relatively simple, and the model becomes proficient at passing the compilation stage.
> * In the **later stages**, the model's CPR gradually stabilizes, while code quality scores show more significant improvement, particularly Topological Similarity (TSS). The change in Geometric Similarity (GS) is less stable, possibly because its feedback signal is less refined and the required learning is more abstract. Additionally, influenced by the step penalty, the average number of interaction rounds required to complete a task slightly decreases.
> * Overall, the model first learns simpler constraints, such as Euler's formula for planar graphs and Maekawa's theorem, before attempting to learn more complex strategies.
>
> We will open-source all our data, environments, training code, models, and complete training details in the short term.
>
> Thank you again for your recognition of our work, as well as for your constructive comments and suggestions!

---

> > ### Comment · Reviewer_VSem · 2025-08-05
> >
> > Thank you for the efforts made to address my concerns. I am quite satisfied with the clarification and explanation about math constraints. Therefore, I would like to raise my score to Accept.

---

> ### Comment · Area_Chair_aVQX · 2025-08-03
>
> Dear Reviewer,
>
> Could you please check if the authors’ rebuttal adequately addresses your concerns? If so, kindly acknowledge the rebuttal and provide any additional comments. If not, it would be greatly appreciated it if you could engage in a discussion with the authors. Your input at this stage is essential to the review process. Thank you very much for your time and effort!
>
> AC

---

### Official Review · Reviewer_DNJx · 2025-07-03

**Clarity:** 3
**Significance:** 4
**Originality:** 3
**Rating:** 5
**Confidence:** 3

**Summary:**

This paper introduces ORIGAMISPACE, a novel benchmark designed to evaluate the complex spatial reasoning capabilities of MLLMs. The authors argue that existing benchmarks fall short in testing multi-step reasoning under precise mathematical constraints. To address this, they use the domain of origami, which naturally involves sequential folding (multi-step reasoning) and adherence to strict geometric principles. The benchmark includes a dataset of 350 origami instances, each with a formal crease pattern (CP), folding process, and final shape. Based on this, they propose four challenging evaluation tasks, including multiple-choice questions and an interactive code generation task. Their evaluation of current MLLMs reveals significant challenges in this domain and demonstrates the potential of interactive and reinforcement learning methods to improve performance.

**Questions:**

1.  The use of origami is highly innovative. Could you elaborate on how you see the specific skills tested here (e.g., understanding crease patterns, mathematical constraints) transferring to other complex spatial reasoning domains like robotic manipulation or architectural design?

2.  The manual collection and verification process for the 350 models must have been very labor-intensive. Have you considered semi-automated methods for expanding the dataset, perhaps by procedurally generating valid CPs and using the compiler to filter them?

3.  The reinforcement learning results are very promising. Could you provide more insight into the policy the agent learns? For example, does it learn to satisfy certain geometric constraints (like Kawasaki's theorem) first before attempting more complex folds?

**Ethical Concerns:**

["NO or VERY MINOR ethics concerns only"]

**Final Justification:**

I thank the authors for their detailed rebuttal. They have provided sufficient answers to my questions regarding the transferability of the skills tested by ORIGAMISPACE, their future plans for dataset expansion, and the specifics of the policy learned by the reinforcement learning agent.
My assessment of this work remains unchanged. The paper's merits—such as its novel use of origami as a testbed for spatial reasoning and the comprehensive design of the benchmark—are clear. The authors' responses have adequately addressed the minor points I raised. Therefore, I will maintain my original positive score.

**Limitations:**

yes

**Paper Formatting Concerns:**

I did not notice any major formatting issues. The paper appears to adhere to the NeurIPS 2025 formatting instructions regarding page limits, style, and structure.

**Quality:**

3

**Strengths And Weaknesses:**

The primary strength of this paper is its highly innovative and well-motivated choice of origami as a domain for evaluating spatial reasoning. This is a significant step beyond existing benchmarks. Origami inherently combines multi-step procedural understanding, 2D-to-3D spatial transformation, and strict, verifiable mathematical constraints (e.g., Kawasaki's and Maekawa's theorems), providing a rich and challenging testbed that current models struggle with. The comprehensive design of the benchmark is another major strength; it is not a single task but a suite of four distinct evaluations that probe different facets of spatial intelligence, from holistic shape prediction to fine-grained generative control.

Furthermore, the technical contribution extends beyond a static dataset. The authors have improved an origami compiler and built an interactive environment for the code generation task. This is a crucial element that elevates the work, as it provides a platform not just for evaluation but for actively training and improving models via environmental feedback and reinforcement learning. The initial results from this RL exploration are very promising and open up a new research direction.

While the work is excellent, a few minor points could be considered. The dataset size of 350 instances is modest, though this is understandable given the high complexity and manual verification required for each origami model, emphasizing quality over quantity. The direct transferability of skills learned on this benchmark to other real-world spatial reasoning domains like robotic manipulation remains an open question for future research, which is a common characteristic of specialized benchmarks.

---

> ### Author Rebuttal · Authors · 2025-07-29
>
> We sincerely appreciate your thorough review and valuable suggestions. We have provided detailed responses to all of your comments and questions. We hope that our clarifications will help you to better understand the specifics of our work and findings.
>
> > The use of origami is highly innovative. Could you elaborate on how you see the specific skills tested here (e.g., understanding crease patterns, mathematical constraints) transferring to other complex spatial reasoning domains like robotic manipulation or architectural design?
>
> We believe that the core competency tested by ORIGAMISPACE—multi-step spatial reasoning under strict constraints—is universal across multiple domains. This ability is transferable because it simulates common spatial challenges. For example:
>
> * In **robotic manipulation**, the sequential folding and mathematical constraints of origami correspond to the multi-step manipulation planning for deformable objects and their physical limitations, respectively.
> * In **architectural design**, predicting a 3D model from a Crease Pattern (CP) is analogous to conceptualizing a building from a 2D blueprint, while the code generation task shares similarities with the parametric design logic in CAD.
>
> To transfer this capability from origami to other domains, we believe the following technical aspects require attention:
>
> *   **Mapping of Representations:** Build a bridge to map problems from the new domain (e.g., point clouds for robotics, CAD files for architecture) into code or symbolic sequences that the model can better understand.
> *   **Generalization of Constraints:** Enable the model to understand and apply equivalent physical or design rules in the new domain (e.g., material non-stretchability, structural load-bearing requirements for buildings). This may involve constructing similarly strict constraints and interactive environments across different fields.
> *   **Unified Action Space:** For generative tasks like robotic manipulation, the "folding" action in origami needs to be abstracted into more universal "manipulation primitives," allowing the model to learn a unified decision-making policy in the new domain.
>
> > The manual collection and verification process for the 350 models must have been very labor-intensive. Have you considered semi-automated methods for expanding the dataset, perhaps by procedurally generating valid CPs and using the compiler to filter them?
>
> Thank you for your valuable suggestion; this is indeed a key focus for our future work.
>
> We did attempt semi-automated methods in the early stages but found that the usability rate of Crease Patterns (CPs) generated by the model was extremely low (less than a 1% success rate) after manual review, while the cost of model generation and human verification was prohibitive. We believe that the capabilities of current Multimodal Large Language Models (MLLMs) are not yet sufficient for high-quality, semi-automated generation tasks. This will likely become feasible only after continued iteration and optimization of model versions in the future.
>
> Therefore, our current strategy is to collaborate with established origami communities and databases. We have reached cooperation intentions with global origami communities and the creators of several origami resource websites, such as the Origami Database. Through official partnerships, we plan to access their larger-scale, manually verified, and curated origami model data. This approach not only ensures the quality and accuracy of the data but also allows us to expand our dataset more efficiently beyond the current 350 instances.
>
> Looking ahead, we see these two approaches as complementary. As MLLM capabilities improve, we will revisit and deploy a semi-automated generation pipeline. We can also involve our community experts in a "human-in-the-loop" process to filter and refine the procedurally generated models, thereby balancing efficiency and quality.
>
> > The reinforcement learning results are very promising. Could you provide more insight into the policy the agent learns? For example, does it learn to satisfy certain geometric constraints (like Kawasaki's theorem) first before attempting more complex folds?
>
> In the early stages of training, the model's compilation success rate (CPR) is low, and the primary feedback is a fixed negative penalty for compilation failures. During the middle phase, rewards related to compilation increase rapidly, especially for avoiding CP Code Syntax Errors (CSE) and Geometrically Impossible Folds (GIF). We believe this is because these two constraints are relatively simple, and the model learns to pass the compilation stage more effectively. In the later stages, the model's CPR gradually stabilizes, while the code quality scores, particularly Topological Similarity (TSS), show more significant improvement. The change in Geometric Similarity (GS) is less stable, possibly because its feedback signal is less refined and the required learning is more abstract. Additionally, influenced by the step penalty, the average number of interaction rounds required to complete a task slightly decreases. Overall, similar to what you suggested, the model first learns simpler constraints, such as Euler's formula for planar graphs and Maekawa's theorem, before attempting to learn more complex strategies.
>
> We will open-source all our data, environments, training code, models, and complete training details in the short term.
>
>
> Thank you again for your recognition of our work, as well as for your constructive comments and suggestions!

---

> > ### Comment · Reviewer_DNJx · 2025-08-06
> >
> > Thanks to the authors for their rebuttals, and I will keep my original positive score.

---

> ### Comment · Area_Chair_aVQX · 2025-08-03
>
> Dear Reviewer,
>
> Could you please check if the authors’ rebuttal adequately addresses your concerns? If so, kindly acknowledge the rebuttal and provide any additional comments. If not, it would be greatly appreciated it if you could engage in a discussion with the authors. Your input at this stage is essential to the review process. Thank you very much for your time and effort!
>
> AC

---

### Decision · Program_Chairs · 2025-09-17

**Decision:**

Accept (spotlight)

**Comment:**

This paper introduces a novel dataset designed to evaluate the ability of multimodal large language models to perform multi-step spatial reasoning. The authors propose several benchmark tasks, including Pattern Prediction, Multi-step Spatial Reasoning, Spatial Relationship Prediction, and End-to-End CP Code Generation, and assess the performance of multiple foundational models on these tasks.

All the reviewers strongly appreciate the creation of the benchmark and agree that it evaluates a uniquely challenging ability of foundation models. Some minor issues were raised by the reviewers, but these were well addressed by the authors during the rebuttal. One only concern is the size of the proposed dataset is small.

Given the strong agreement among the reviewers, AC recommends an accept for this paper.